# Reweighting Local Mimina with Tilted SAM

## Abstract

Sharpness-Aware Minimization (SAM) has been demonstrated to improve the generalization performance of overparameterized models by seeking flat minima on the loss landscape through optimizing model parameters that incur the largest loss within a neighborhood. Nevertheless, such min-max formulations are computationally challenging especially when the problem is highly non-convex. Additionally, focusing only on the worst-case local solution while ignoring potentially many other local solutions may be suboptimal when searching for flat minima. In this work, we propose Tilted SAM (TSAM), a generalization of SAM inspired by exponential tilting that effectively assigns higher priority to local solutions that are flatter and that incur larger losses. TSAM is parameterized by a tilt hyperparameter $t$ and reduces to SAM as $t$ approaches infinity. We prove that (1) the TSAM objective is smoother than SAM and thus easier to optimize; and (2) TSAM explicitly favors flatter minima as $t$ increases. This is desirable as flatter minima could have better generalization properties for certain tasks. We develop algorithms motivated by the discretization of Hamiltonian dynamics to solve TSAM. Empirically, TSAM arrives at flatter local minima and results in superior test performance than the baselines of SAM and ERM across a range of image and text tasks.

## 1 Introduction

Empirical risk minimization (ERM) is a classic framework for machine learning that optimizes for the average performance of the observed samples. For $n$ training samples $\{x_i\}_{i \in [n]}$ (which may also contain label information), model parameters $\theta \in \mathbb{R}^d$, and a loss function $l(\cdot)$, let ERM be defined as

$$\min_\theta L(\theta) := \frac{1}{n} \sum_{i \in [n]} l(x_i; \theta). \tag{1}$$

In overparameterized models, however, minimizing ERM may arrive at a bad local minimum. To address this, one line of work focuses on minimizing the sharpness of final solutions, ensuring that the losses of parameters around local minima are uniformly small. One popular formulation is sharpness-aware minimization (SAM), that optimizes over the worst-case loss over perturbed parameters. For a perturbing region $\|\epsilon\| \leqslant \rho$ where $\epsilon \in \mathbb{R}^d$, the SAM objective is defined as

$$\min_\theta L^s(\theta) := \max_{\|\epsilon\| \leqslant \rho} L(\theta + \epsilon). \tag{2}$$

Typically, SAM is optimized by alternating between running gradient ascent (to find the max loss) and gradient descent steps (to minimize the max loss) on model parameters. However, it is difficult for such updating steps (and its variants) to find the exact perturbation $\epsilon$ that incurs the true max loss, as the loss landscape can be highly non-convex and potentially non-smooth. In addition, ignoring potentially many other large-loss regions may still leave some areas of the loss surface sharp. For instance, we have computed the average loss across the neighborhoods of SAM solutions, and find that it is still higher than the ones obtained by our approach (Section 5.2).

To this end, we propose a generalized and smoothed variant of SAM inspired by exponential tilting and its widespread usage in probability and statistics. In optimization literature, it has also been used as an efficient min-max smoothing operator (Kort & Bertsekas, 1972). Tilted SAM (TSAM), parameterized by a tilt scalar $t \geqslant 0$, is defined as

$$\min_\theta L^t(\theta) := \frac{1}{t} \log \left( \int e^{tL(\theta+\epsilon)} d\mu(\epsilon) \right) = \frac{1}{t} \log \left( \mathbb{E}_{\mu(\epsilon)} \left[ e^{tL(\theta+\epsilon)} \right] \right), \tag{3}$$

where $L(\theta + \epsilon)$ is defined in Eq. (1), $\mu(\epsilon)$ denotes an uncertainty probability measure for $\epsilon$ that can represent uniform balls such as $\|\epsilon\| \leqslant \rho$ (but other measures are possible as well), and $\epsilon \in \mathbb{R}^d$. When $t \to \infty$ and $\mu(\epsilon)$ takes $\|\epsilon\| \leqslant \rho$, $L^t(\theta)$ reduces to the SAM objective $L^s(\theta)$. When $t = 0$, $L^t(\theta)$ reduces to the average loss over the perturbed neighborhood $\mu(\epsilon)$, i.e., $\mathbb{E}_{\mu(\epsilon)}[L(\theta + \epsilon)]$ where the expectation is taken with respect to the randomness of $\epsilon$ (formally proven in Appendix B). When both $t = 0$ and $\rho = 0$, the loss is reduced just to the classic average empirical risk $L(\theta)$. We use $\mathbb{E}_{\mu(\epsilon)}$, $\mathbb{E}_\epsilon$, and $\mathbb{E}$ interchangeably when the meaning is clear from the context.

TSAM provides a smooth transition between min-max optimization (Eq. (2)) and min-avg optimization $\min_\theta \mathbb{E}_{\mu(\epsilon)}[L(\theta + \epsilon)]$. The min-avg optimization has appeared in prior works known as average-perturbed sharpness (Wen et al., 2022), noise-perturbed loss (Zhang et al., 2024), or random smoothing (Duchi et al., 2012). The smoothness parameter of the TSAM objective increases as the value of $t$ increases, which suggests that it is easier to optimize than SAM (Section 3). As we formalize later, TSAM reweights gradients of neighboring solutions based on their loss values, which can be viewed as a soft version of SAM which assigns all the weights to one single worst minimum. In addition to the benefits in optimization, rigorously considering many, as opposed to one, neighbourhood parameters that incur large losses results in improved generalization. We provide both theoretical characterization and empirical evidence showing that TSAM solutions are flatter than those of ERM and SAM. One line of the mostly related works have explored tilted risks to reweight different data points (Li et al., 2023; Robey et al., 2022). In this work, we use the TSAM framework to assign varying priority to local minima in the parameter space.

To solve TSAM, we need to estimate the integral over $\mu(\epsilon)$ (Eq. (3)), or equivalently, to estimate the full gradient of the objective, which is a tilted aggregation of gradients evaluated at $L(\theta + \epsilon)$. Both require sampling the perturbation $\epsilon$ with probability proportional to $e^{tL(\theta+\epsilon)}$ for the integration. Naively sampling $\epsilon$ at random to obtain $L^t(\theta)$ would be inefficient, as it is likely that $L(\theta + \epsilon)$ under the sampled $\epsilon$ is small and therefore we need many samples to converge to the true distribution. On the other hand, methods based on Hamiltonian Monte Carlo (HMC) (Leimkuhler & Reich, 2004) are more principled and guaranteed to arrive at the exact distribution. Inspired by the Euler's rules for HMC, we develop an algorithm to efficiently sample $\epsilon$'s and estimate the true gradient of $L^t(\theta)$.

**Contributions.** We propose TSAM, a new optimization objective that reweights the parameters around local minima via exponential tilting. We rigorously study several properties of TSAM, showing that it always favors flatter solutions as $t$ increases and achieves a tighter generalization bound than SAM for modest values of $t$ (Section 3). To optimize TSAM, we adapt a specific HMC algorithm to efficiently sample the model perturbation $\epsilon$ (Section 4). We empirically demonstrate that TSAM results in flatter solutions and superior generalization performance than SAM and its variants for deep neural networks including transformers on both image and text datasets (Section 5).

## 2 RELATED WORK

**Sharpness-Aware Minimization.** SAM regularizes overparameterized models by considering adversarial data points that have large training errors (Foret et al., 2020; Zheng et al., 2021). The SAM variants, training dynamics, and applications in different models have been extensively studied in prior work (Andriushchenko & Flammarion, 2022; Baek et al., 2024; Bartlett et al., 2023; Chen et al., 2021; 2024; Du et al., 2022; Foret et al., 2020; Kwon et al., 2021; Liu et al., 2022b; Long & Bartlett, 2023; Mi et al., 2022; Mueller et al., 2023; Xie et al., 2024; Zhao et al., 2022; Zhou et al., 2021; Zhuang et al., 2022). Some work aim to improve efficiency of the SAM algorithm studying different relaxations (Du et al., 2022; Liu et al., 2022a). Zhao et al. (2022) use a linear interpolation between normal gradients and SAM outer gradients evaluated at the max-loss parameter, which does not take into account the possibly many bad local minima for highly non-convex problems. Liu et al. (2022b) improve the inner max optimization by adding a random perturbation to the gradient ascent step to smoothen its trajectory. Li & Giannakis (2023) leverages a moving average of stochastic gradients in the ascent direction to reduce the gradient variance[1]. Our goal is *not* to better approximate the inner max or develop algorithms for solving the min-max SAM formulation, but rather, to solve a different TSAM objective that reweights many local minima to seek flat solutions. Nevertheless,

---

[1]Note that the notion of 'variance' in VASSO (Li & Giannakis, 2023) refers to the variance of stochastic gradients compared with full gradients; whereas in TSAM, we examine the loss variance around the neighborhood regions, where the randomness in Definition 2 comes from the perturbation $\epsilon$.

we still compare with more advanced algorithms for the SAM objective (Section 5) and show the superiority of TSAM solutions. Zhou et al. (2021) perform sample-wise reweighting for SAM, as opposed to parameter-wise reweighting proposed herein. As TSAM is a new objective, in principle, we can readily apply many existing optimization techniques (that can be potentially applied to SAM as well) such as variance reduction (Johnson & Zhang, 2013), acceleration (Nesterov, 1983), or adaptivity (Duchi et al., 2011; Kingma & Ba, 2014; Streeter & McMahan, 2010) on top of the tilted stochastic gradients to gain further improvement.

There is also work attempting to understand why SAM leads to better generalization or theoretically characterize what SAM (and its implementation) is effectively minimizing (Andriushchenko & Flammarion, 2022; Chen et al., 2024; Long & Bartlett, 2023; Wen et al., 2022). In this work, we prove that TSAM (and SAM) encourages flatter models for a class of problems including generalized linear models, where flatness (or sharpness) by the variance of the losses around the minima (Definition 3). Our proposed TSAM framework is particularly suitable for problems where flatness helps generalization. The various notions of sharpness, along with theoretical relations between sharpness and generalization still remain an open problem (Andriushchenko et al., 2023; Ding et al., 2024; Wen et al., 2024), which is outside the scope of our paper.

**Tilting in Machine Learning.** Exponential tilting, used to shift parametric distributions, has appeared in previous literature in importance sampling, optimization, and information theory (e.g., Aminian et al., 2024; Dembo, 2009; Kort & Bertsekas, 1972; Siegmund, 1976). Recently, the idea of tilted risk minimization (which exponentially reweights different training samples) has been explored in machine learning applications such as enforcing fairness and robustness, image segmentation, and noisy label correction (Aminian et al., 2024; Li et al., 2023; Robey et al., 2022; Szabó et al., 2021; Zhou et al., 2020). A closely-related LogSumExp operator is often used to as an smooth approximation to the max, which is always considered more computationally favorable (Calafiore & El Ghaoui, 2014; Kort & Bertsekas, 1972; Li et al., 2023; Shen & Li, 2010). One application of tilted risks applied to the adversarial training problem is to balance worst-case robustness (i.e., adversarial robustness) and average-case robustness in the data space (Robey et al., 2022), among other approaches that can also achieve a transition between worst-case and average-case errors (Rice et al., 2021). Our work is similar conceptually, but we consider reweighting adversarial model parameters, instead of adversarial data points. Compared with SAM (optimizing the largest loss), the TSAM framework offers additional flexibility of optimizing over quantiles of losses given the connections between tilting and quantile approaches (Li et al., 2023; Rockafellar et al., 2000).

## 3 PROPERTIES OF TSAM

In this section, we discuss properties of the TSAM objective. We first state the convexity and smoothness of TSAM (Section 3.1). We then show that as $t$ increases, the gap between less-flat and more-flat solutions measured in terms of the TSAM objective becomes larger. In other words, optimizing TSAM would give a flatter solution as $t$ increases (Section 3.2). Finally, we discuss the generalization behavior of TSAM and prove that there exists $t \in (0, \infty)$ that result in the tightest generalization bound (Section 3.3). All properties discussed in this section hold regardless of the distributions of $\epsilon$ (i.e., choice of $\mu(\epsilon)$), unless otherwise specified.

### 3.1 CONVEXITY AND SMOOTHNESS

In this part, we connect the convexity and smoothness of TSAM with the convexity and smoothness of the ERM loss. We provide complete proofs in Appendix B. We first define a useful quantity (tilted weights) that will be used throughout this section.

**Definition 1** ($t$-tilted weights). *For a perturbed model parameter $\theta + \epsilon$, we define its corresponding $t$-tilted weight as $w^t(\theta + \epsilon) := \frac{e^{tL(\theta+\epsilon)}}{\mathbb{E}[e^{tL(\theta+\epsilon)}]}$.*

This is exponentially proportional to the loss evaluated at parameter values $\theta + \epsilon$. The expectation is with respect to the randomness of $\epsilon$ constrained by $\mu(\epsilon)$. When $t = 0$, 0-tilted weights are uniform. When $t \to \infty$, $w^t(\theta + \epsilon)$ focuses on the max loss among all possible $\{\theta + \epsilon\}$. Such weights have appeared in previous literature on importance sampling (Siegmund, 1976), but they are only applied to reweight sample-specific losses, as opposed to perturbation-specific parameters. Given tilted weights in Definition 1, we can present the TSAM gradients and Hessian as follows.

**Lemma 1** (Gradient and Hessian for TSAM). *Assume $L(\cdot)$ is continuously differentiable. The full gradient of TSAM (Objective (3)) is*

$$\nabla L^t(\theta) = \frac{\mathbb{E}[e^{tL(\theta+\epsilon)}\nabla L(\theta+\epsilon)]}{\mathbb{E}[e^{tL(\theta+\epsilon)}]} = \mathbb{E}[w^t(\theta+\epsilon)\nabla L(\theta+\epsilon)]. \tag{4}$$

*The Hessian of TSAM $\nabla^2 L^t(\theta)$ is*

$$t\left(\mathbb{E}\left[w^t(\theta+\epsilon)\nabla L(\theta+\epsilon)^\top \nabla L(\theta+\epsilon)\right] - \mathbb{E}\left[w^t(\theta+\epsilon)\nabla L(\theta+\epsilon)\right]^\top \mathbb{E}\left[w^t(\theta+\epsilon)\nabla L(\theta+\epsilon)\right]\right)$$
$$+ \mathbb{E}\left[w^t(\theta+\epsilon)\nabla^2 L(\theta+\epsilon)\right]. \tag{5}$$

The gradient of TSAM can be viewed as reweighting the gradients of $\nabla L(\theta+\epsilon)$ by the loss values $e^{tL(\theta+\epsilon)}$. Examining the Hessian, we note that the first term is $t$ multiplied by a positive semi-definite matrix, and the second term can be viewed as a reweighting of the Hessian of the original loss $L$.

It is not difficult to observe that if $L(\theta)$ is $p$-Lipschitz with respect to $\theta$, then $L^t(\theta)$ is $p$-Lipschitz with respect to $\theta$. If $L$ is $\mu$-strongly convex, then $L^t$ is also $\mu$-strongly convex (proved in Appendix B). Next, we show that the smoothness of TSAM scales linearly with $t$.

**Lemma 2** (Smoothness of TSAM). *Let $L(\cdot)$ be $\beta$-smooth and $\beta$ is bounded. Then $L^t(\theta)$ is $\beta(t)$-smooth, where $\beta(t)$ satisfies $0 < \lim_{t\to\infty}\frac{\beta(t)}{t} < +\infty$.*

That is, $\beta(t) = O(t)$. The proof is deferred to Appendix B. In Lemma 2, we connect the smoothness of the tilted objective with the smoothness of the original ERM objective. We see that for any bounded $t$, the smoothness parameter is bounded. As $t$ increases, TSAM becomes more difficult to optimize as the loss becomes more and more non-smooth. When $t \to \infty$, $\beta(t) \to \infty$. If we have access to unbiased gradient estimates at each round, it directly follows that the convergence of SAM (TSAM with $t \to \infty$) objective is slower than that of tilted SAM following standard arguments (Nesterov, 2013). To further visualize this, we create a one-dimensional toy problem in Appendix A where we obtain the globally optimal solutions for each objective. We show that both SAM and TSAM are able to arrive at a flat solution; but the SAM objective is non-smooth, hence more difficult to optimize.

## 3.2 TSAM PREFERS FLATTER MODELS AS $t$ INCREASES

In this subsection, we focus on a specific class of models including generalized linear models (GLMs), where the loss function $l(x_i;\theta)$ carries the form of

$$l(x_i;\theta) = A(\theta) - \theta^\top T(x_i), \quad L(\theta) = A(\theta) - \theta^\top\left(\frac{1}{n}\sum_{i\in[n]}T(x_i)\right) := A(\theta) - \theta^\top T(x). \tag{6}$$

For GLMs, $A(\theta)$ is a convex function, and $\sum_{i\in[n]} T(x_i)T(x_i)^\top \succ 0$ (Wainwright & Jordan, 2008). Our results in this section apply to loss functions defined in Eq. (6), which subsume linear models. Before introducing the main theorem, we define two important quantities that will be used throughout this section, and in the experiments.

**Definition 2** ($t$-weighted mean and variance). *We define $t$-weighted mean of a random variable $X$ $\frac{\mathbb{E}[e^{tX}X]}{\mathbb{E}[e^{tX}]}$. Similarly, we define $t$-weighted variance of a random variable $X$ as $\frac{\mathbb{E}[e^{tX}X^2]}{\mathbb{E}[e^{tX}]} - \left(\frac{\mathbb{E}[e^{tX}X]}{\mathbb{E}[e^{tX}]}\right)^2$.*

When $t = 0$, these definitions reduce to standard mean $\mathbb{E}[X]$ and variance $\mathbb{E}[X^2] - (\mathbb{E}[X])^2$. Similar tilted statistics definitions have also appeared in prior work (Li et al., 2023). We leverage weighted variance to define sharpness below.

**Definition 3** ($t$-sharpness). *We say that a model parameter is $\theta_1$ is $t$-sharper than $\theta_2$ if the $t$-weighted variance of $L(\theta_1+\epsilon)$ (which is a random variable for loss distribution under model parameters perturbed by $\epsilon$) is larger than the $t$-weighted variance of $L(\theta_2+\epsilon)$.*

Given the definition of sharpness above based on weighted variance, we are ready to prove that TSAM encourages flatter local minima as the increase of $t$. Empirically, in Section 5, we also plot the 0-sharpness of the solutions obtained from different objectives, and observe that TSAM achieves smaller sharpness values (measured in terms of standard variance) than ERM and SAM. Proper definitions of sharpness is generally still an open problem, and other options are possible such as the trace of Hessian and gradient norms (Wen et al., 2022).

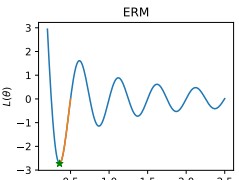 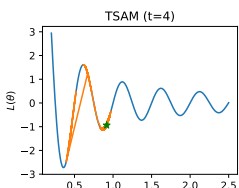 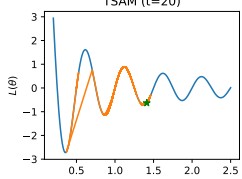 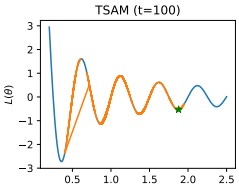

Figure 1: Optimization trajectories of different objectives (orange) with final solutions marked in star. The local minima get flatter from left to right in each subfigure. We see that TSAM favors flat minima as $t$ increases. ERM solution by gradient descent converges to a sharp minimal. We solve TSAM on this one-dimensional problem by sampling thousands of $\epsilon$'s to estimate the gradient of $L^t(\theta)$ (Eq. (4)) at each step. This is not feasible for real problems.

**Theorem 1** (TSAM prefers flatter models as $t$ increases). *Assume $L(\theta)$ is given by Eq. (6) and $L(\theta)$ is continuously differentiable. For any $\theta_1, \theta_2 \in \mathbb{R}^d$, let $g^t(\theta_1, \theta_2) := L^t(\theta_1) - L^t(\theta_2)$. If $\theta_1$ is $t$-sharper than $\theta_2$, then $\frac{\partial g^t(\theta_1, \theta_2)}{\partial t} \geqslant 0$.*

For some $\theta_1$ sharper than $\theta_2$, it is possible that $L(\theta_1) = L(\theta_2)$, which implies that ERM is not able to distinguish between the two solutions, while TSAM can. Furthermore, Theorem 1 indicates that as $t$ increases, the TSAM objective favors $\theta_2$ more aggressively, as the gap between $L^t(\theta_1)$ and $L^t(\theta)$ grows larger. We explore a one-dimensional toy problem with many local minima: $L(\theta) = 2\sin(4\pi\theta)/(2\theta) + 0.005(\theta - 1)^2$, and focus on the area $\theta \in (0.2, 2.5)$ to visualize this behavior. We take $\rho = 0.2$ and a fixed learning rate $0.005$ for all objectives. Each run starts from the same initialization $\theta = 0.5$. In Figure 1, we see that as $t$ increases, TSAM leads to flatter solutions, despite having larger objective values measured by $L(\theta)$. As a side note, we prove that for any $\theta$, the objective value of $L^t(\theta)$ is monotonically increasing as $t$ increases (Appendix B). Next, we discuss a special case when $t$ is close to 0, where we provide another perspective on the TSAM behavior.

**Discussions for $t \to 0$.** All the results above hold for the small-$t$ regime, where sharpness reduces to standard variance when $t \to 0$ (Definition 2). It still follows that $\left.\frac{\partial g^t(\theta_1, \theta_2)}{\partial t}\right|_{t \to 0} \geqslant 0$ if $\theta_1$ is sharper than $\theta_2$. Here, we provide another interpretation of TSAM when $t$ is close to zero. Similar statements have also appeared in prior works in a different context (e.g., Li et al., 2023; Liu & Theodorou, 2019). For a very small $t$, it holds that $\frac{1}{t} \log \left( \mathbb{E}\left[ e^{tL(\theta + \epsilon)} \right] \right) \approx \mathbb{E}[L(\theta + \epsilon)] + \frac{t}{2} \mathrm{var}\left( L(\theta + \epsilon) \right) + o(t^2)$. We provide a proof in Appendix B. Hence, optimizing TSAM is approximately equivalent to optimizing for the mean plus variance of the losses under the perturbed parameters. When $t = 0$, it reduces to only optimizing for $\mathbb{E}[L(\theta + \epsilon)]$. In other words, TSAM with $t$ close to 0 is directly minimizing 0-sharpness (standard variance). For any $\theta_1$ and $\theta_2$ such that $\theta_1$ is sharper than $\theta_2$, we have

$$g^t(\theta_1, \theta_2) \approx \mathbb{E}[L(\theta_1 + \epsilon)] + \frac{t}{2}\mathrm{var}\left( L(\theta_1 + \epsilon) \right) - \mathbb{E}[L(\theta_2 + \epsilon)] - \frac{t}{2}\mathrm{var}\left( L(\theta_2 + \epsilon) \right), \quad (7)$$

$$\frac{\partial g^t(\theta_1, \theta_2)}{\partial t} \approx \frac{1}{2}\left( \mathrm{var}\left( L(\theta_1 + \epsilon) \right) - \mathrm{var}\left( L(\theta_2 + \epsilon) \right) \right) \geqslant 0. \quad (8)$$

This is a special case of Theorem 1 for $t \to 0$. It suggests that as we increase $t$ from 0 for a small amount, the standard variance of neighborhood loss would reduce. Note that some recent works propose a noise-injected loss similar to TSAM with $t \to 0$ (Zhang et al., 2024). The proposed algorithm therein explicitly minimizes the trace of Hessian, which aligns with our arguments that the TSAM objective can lead to flatter solutions.

So far, we study properties of TSAM regarding convexity, smoothness of the objective, and sharpness of the resulting solutions. We note that these properties of the objective are independent of the actual optimization algorithms used to optimize TSAM. Though Theorem 1 implies similar benefits of both TSAM and SAM relative to ERM (assuming we optimize TSAM and SAM perfectly), Lemma 2 shows the superiority of the TSAM objective over SAM with unbounded smoothness parameters, as TSAM is easier to optimize. The performance of practical applications depends on both the properties of the objectives and the approximation algorithms used to solve them.

### 3.3 GENERALIZATION OF TSAM

In this section, we give a uniform bound on the generalization error of our TSAM objective. By solving the tilted objective empirically, during test time, we are ultimately interested in evaluating the *linear*

*population risk* $\mathbb{E}_Z[l(\theta; Z)]$ where $Z$ denotes the underlying data distribution and $\{x_i\}_{i \in [n]} \sim Z$. We define generalization error as the difference between population risk and our empirical objective value $\mathbb{E}_Z[l(\theta; Z)] - \frac{1}{t} \log \mathbb{E}_\epsilon[e^{tL(\theta+\epsilon)}]$, bounded as follows.

**Theorem 2** (Generalization of TSAM). *Assume losses are bounded as $0 \leqslant L(\cdot) \leqslant M$. Suppose we have $n$ training data points. For any $\theta \in \Theta$ and $t \geqslant 0$, with probability $1 - \delta$, the difference between population risk and empirical TSAM risk satisfies*

$$\mathbb{E}_Z[l(\theta; Z)] - \frac{1}{t} \log \mathbb{E}_\epsilon[e^{tL(\theta+\epsilon)}] \leqslant M\sqrt{\frac{\log(2/\delta)}{2n}} - \frac{\mathrm{var}_\epsilon(e^{tL(\theta+\epsilon)})}{2te^{2tM}} + c, \qquad (9)$$

*where $c := L(\theta) - \mathbb{E}_\epsilon[L(\theta + \epsilon)]$ is a constant independent of $t$.*

In other words, we have for any $\theta$, with probability $1 - \delta$,

$$\mathbb{E}_Z[l(\theta; Z)] \leqslant \frac{1}{t} \log \mathbb{E}_\epsilon[e^{tL(\theta+\epsilon)}] + M\sqrt{\frac{\log(2/\delta)}{2n}} - \frac{\mathrm{var}_\epsilon(e^{tL(\theta+\epsilon)})}{2te^{2tM}} + c. \qquad (10)$$

We defer the proof to Appendix B, where we build upon existing generalization results of a related objective (Aminian et al., 2024). From Theorem 2, we see that when the sample space of $\epsilon$ is empty, our result reduces to $\mathbb{E}_Z[l(\theta, Z)] \leqslant L(\theta) + M\sqrt{\frac{\log(2/\delta)}{2n}}$, scaling at a rate of $\frac{1}{\sqrt{n}}$ consistent with standard uniform bound on the average risk (Shalev-Shwartz & Ben-David, 2014). When $t \to \infty$ and we define $\mu(\epsilon)$ to be $\|\epsilon\| \leqslant \rho$ over some distribution, the result gives an upper bound on the generalization of SAM: $\mathbb{E}_Z[l(\theta, Z)] - \max_{\|\epsilon\| \leqslant \rho} L(\theta + \epsilon) \leqslant M\sqrt{\frac{\log(2/\delta)}{2n}} + c$. For the most interesting case of $t \in (0, \infty)$, we give a remark on the tightness of the bound below.

**Remark 1.** *Denote $\theta^{TSAM}$ and $\theta^{ERM}$ as optimal solutions for TSAM (Eq. (3)) and ERM (Eq. (1)), respectively. For modest values of $t$, due to the negativity of $-\frac{\mathrm{var}_\epsilon(e^{tL(\theta+\epsilon)})}{2te^{2tM}}$, the upper bound of the linear population risk $\mathbb{E}_Z[l(\theta^{TSAM}, Z)]$ (right-hend side of Eq. (10)) can be smaller than that of the linear risk $\mathbb{E}_Z[l(\theta^{ERM}, Z)]$, as long as $\frac{1}{t} \log \mathbb{E}_\epsilon[e^{tL(\theta^{TSAM}+\epsilon)}] - \frac{\mathrm{var}_\epsilon(e^{tL(\theta^{TSAM}+\epsilon)})}{2te^{2tM}} \leqslant L(\theta^{ERM})$. This implies that by solving TSAM, we can obtain a solution that results in a smaller upper bound of the linear population error than that of ERM.*

## 4 ALGORITHMS

In this section, we describe the algorithms we use to solve TSAM. The main challenge in solving TSAM is to sample $\epsilon$ to get a good estimator of $L^t(\theta)$, or equivalently, $\nabla L^t(\theta)$. We first describe a general approach where we use estimated tilted gradients (given sampled $\epsilon$'s) to update the model (Section 4.1). Then, we discuss how to sample $\epsilon$'s via a specific Hamiltonian Monte Carlo algorithm and present our method and implementation (Section 4.2).

### 4.1 GENERAL ALGORITHM

To solve TSAM, the primary challenge is to estimate the integral $\frac{1}{t} \log \left( \int e^{tL(\theta+\epsilon)} d\mu(\epsilon) \right)$, or its full gradient $\frac{\mathbb{E}[e^{tL(\theta+\epsilon)}\nabla L(\theta+\epsilon)]}{\mathbb{E}[e^{tL(\theta+\epsilon)}]}$, assuming gradient-based methods and the differentiable loss $L$. A naive way is to first sample $\epsilon$ from $\mu(\epsilon)$ following the pre-defined distribution (e.g., Gaussian or uniform) over $\mu(\epsilon)$, and then perform tilted aggregation with weights proportional to $e^{tL(\theta+\epsilon)}$. However, this approach may be extremely inefficient, as there could be an infinite set of perturbed model parameters with relatively small losses, which are not informative. In Figure 8 in the appendix, we empirically show that even when we sample a much larger number of $\epsilon$'s, the resulting accuracy is still worse than our proposed method. Instead, we propose to sample $s$ number of $\epsilon$'s from distribution $e^{\delta L(\theta+\epsilon)}$ (denoted as $\{\epsilon_j\}_{j \in [s]}$), where $0 \leqslant \delta \leqslant t$. We then use these $\{\epsilon_j\}_{j \in [s]}$ to obtain an empirical gradient estimation with weights proportional to $\{e^{(t-\delta)L(\theta+\epsilon_j)}\}_{j \in [s]}$, as the full gradient is a tilted average of the original gradient on $L(\cdot)$. To improve sample efficiency, we use gradient-based methods such as Hamiltonian Monte Carlo (HMC) that simulates Hamiltonian dynamics (Leimkuhler & Reich, 2004). The structure of our proposed method is in Algorithm 1. Note that in principle, after estimating the tilted stochastic gradients, we can further apply existing optimization techniques such as variance reduction (Johnson & Zhang, 2013), acceleration (Nesterov), or adaptivity (Duchi et al., 2011; Streeter & McMahan, 2010) to gain further improvement, which we leave for future work.

---

**Algorithm 1:** Tilted SAM Solver

---

**Input:** $t$, $\theta^0$, learning rate $\eta$, total iterations $T$, total number of samples $s$

**for** $i = 0, \cdots, T-1$ **do**

    Sample $s$ random perturbations $\{\epsilon_j\}_{j\in[s]}$ from distribution $e^{\delta L(\theta^i+\epsilon)}$ under the constraint
    characterized by $\mu(\epsilon)$ via some HMC algorithm (Algorithm 2)

    Update $\theta^i$ with the estimated gradient evaluated on the mini-batch:

$$\theta^{i+1} \leftarrow \theta^i - \eta \frac{\sum_{j\in[s]} e^{(t-\delta)L(\theta^i+\epsilon_j)}\nabla L(\theta^i+\epsilon_j)}{\sum_{j\in[s]} e^{(t-\delta)L(\theta^i+\epsilon_j)}}$$

**end**

**return** $\theta^T$

---

### 4.2 SAMPLING $\epsilon$

There could be potentially different algorithms for sampling $\epsilon$ where $p(\epsilon) \propto e^{\delta L(\theta+\epsilon)}$. Here we propose an approximate and cheap sampler based on discretization of Hamiltonian dynamics. Our method is inspired by one of the best-known way to approximate the solution to a system of differential equations, i.e., Euler's method or its modification (Neal et al., 2011). A more accurate solver like the leap-frog method might be more popular for HMC, but these come at an increased expense (Neal et al., 2011). As our goal to minimize computational cost, we stick with the cheaper Euler's approach as follows. We first initialize $\epsilon_0$ from an $L_2$ ball that satisfies $\|\epsilon\| \leqslant \rho$, and initialize the momentum $p_0 \in \mathbb{R}^d$ from some Gaussian distribution, i.e., $p_0 \sim \mathcal{N}(0, \sigma^2\mathbf{I})$. Note that the negative log probability density of the energy function $U(\epsilon)$ is $-\log(e^{\delta L(\theta+\epsilon)}) = -\delta L(\theta+\epsilon)$. At each sampling step, we run the following steps for $N$ iterations with a small step-size $\beta$ to obtain a candidate $\epsilon$:

$$p \leftarrow p + \beta\delta\nabla_\epsilon L(\theta+\epsilon), \quad \epsilon \leftarrow \epsilon + \beta p/\sigma^2. \tag{11}$$

After obtaining a candidate $\epsilon$, we accept $\epsilon$ with probability $\min\{1, e^{\delta L(\theta+\epsilon)-\frac{\|p\|^2}{2\sigma^2}}/e^{\delta L(\theta+\epsilon_0)-\frac{\|p_0\|^2}{2\sigma^2}}\}$. If the candidate $\epsilon$ is not accepted, we set $(p,\epsilon)$ to the initial point before the $N$ iterations. Repeating the above for enough times would give us a sample $\epsilon$ from the exact distribution.

---

**Algorithm 2:** Sampling from $e^{\delta L(\theta^i+\epsilon)}$ under the constraint $\|\epsilon\| \leqslant \rho$

---

**Input:** $\theta^0$, total samples $s$, uncertainty ball radius $\rho$

**for** $j = 0, \cdots, s$ **do**

    Perturb $\theta^i$ with a random $\delta_j$ sampled from Gaussian or uniform distribution: $\theta^i_j \leftarrow \theta^i + \delta_j$

    Run normalized SGD on the mini-batch data at $\theta^i_j$: $\hat{\theta}^i_j \leftarrow \theta^i_j + \rho\frac{\nabla L(\theta^i_j)}{\|\nabla L(\theta^i_j)\|}$;    $\epsilon_j \leftarrow \hat{\theta}^i_j - \theta^i$

**end**

**return** $\{\epsilon_j\}_{j\in[s]}$

---

Generating one $\epsilon$ via HMC requires at least $2N$ gradient evaluations, which is infeasible for large-scale problems. Hence, we set $N = 1$ in all the main experiments, and meanwhile accept the generated $\epsilon$ with probability 1. For completeness, we evaluate the effects of increasing $N$ in HMC in Appendix C. We observe that using $N > 1$ does not significantly improve the performance. Running equations in Eq. (11) for one step, if $p$ is initialized as $p = \mathbf{0}$, we have $\epsilon \leftarrow \epsilon + \beta'\nabla L(\theta+\epsilon)$, where $\beta'$ is a constant. We adapt this updating rule to our problem, and run the aforementioned procedure in parallel for $s$ times to get $s$ samples. Our method is presented in Algorithm 2. Though Algorithm 2 does not guarantee the $\epsilon_j$'s result in a consistent estimator of the TSAM integral, we empirically showcase its effectiveness on non-convex models including transformers in the next section.

## 5 EXPERIMENTS

In this section, we first describe our setup. Then we present our main results, comparing TSAM with the baselines of ERM (Eq. (1)), SAM (Eq. (2)), and SAM variants on both image and text data (Section 5.1). We explain TSAM's superior performance by empirically examining the flatness of local minima in Section 5.2. In Section 5.3, we discuss the effects of hyperparameters.

**Tasks and Datasets.** We consider image tasks involving convolutional neural networks and transformers and the GLUE benchmark of language modeling (Wang, 2018). First, we explore standard training of ResNet18 (He et al., 2016) and WideResNet16-8 (Zagoruyko, 2016) on classification over CIFAR100 (Krizhevsky et al., 2009). Since vision transformers (ViTs) (Dosovitskiy et al., 2020) have been shown to have much sharper local minima than CNNs (Chen et al., 2021), we study the performance on TSAM finetuning ViTs (pretrained on ImageNet (Deng et al., 2009)) on an out-of-distribution Describable Texture Dataset (DTD) (Cimpoi et al., 2014), where the task is 47-class classification. Additionally, previous works show that SAM is robust to label noise (Baek et al., 2024; Foret et al., 2020); and we evaluate in the setting of training ResNet18 on CIFAR100 with uniform label noise generated by substituting 20% of the true labels uniformly at random to other labels. We also use WideResNet to train DTD and noisy CIFAR100 datasets from scratch. Lastly, we study finetuning a pretrained DistilBert (Sanh, 2019) model on the GLUE benchmark including both classification and regression problems on text data. All the experiments are conducted on V100, L40S, or A100 GPUs.

**Hyperparameter Tuning.** We take $\mu(\epsilon)$ to be $\|\epsilon\| \leqslant \rho$ for all TSAM experiments, and tune the $\rho$ parameters separately from $\{0.05, 0.1, 0.2\}$ for relevant methods. For TSAM, we tune $t$ from $\{0, 1, 5, 20, 100\}$ and select the best one based on the validation set. We also report the performance for all $t$'s in the next sections. We use $s=3$ or $s=5$ sampled $\epsilon$'s for all datasets and find that it works well. For some SAM variants that introduce additional hyperparameters, we tune those via grid search as well. We fix the batch size to be 64 for all the datasets and methods, and use constant learning rates tuned from $\{0.0003, 0.001, 0.003, 0.01, 0.03, 0.1\}$ for each algorithm. Despite the existence of adaptive methods for SGD and SAM (Kingma & Ba, 2014; Kwon et al., 2021), we do not incorporate adaptivity for any algorithm for a fair comparison. See Appendix C for details on hyperparameter tuning.

## 5.1 TSAM LEADS TO BETTER TEST PERFORMANCE

We compare the performance of various objectives and algorithms in Table 1. ERM denotes minimizing the empirical average loss with mini-batch SGD. SAM is the vanilla SAM implementation with one step of gradient ascent and one step of gradient descent at each iteration (Foret et al., 2020). Note that TSAM requires more gradient evaluations per iteration. Hence, we include two additional baselines of SAM **under the same computational budget as TSAM** runs. (1) We simply run the vanilla SAM algorithm for more iterations until it reaches the same runtime as TSAM. (2) We try another SAM approximation by exploring different step sizes along the gradient ascent directions and pick the one incurring the biggest loss. Then we evaluate the gradient under that step size to be applied to the original model parameters. We call these expansive SAM baselines ESAM1, and ESAM2, respectively. We also evaluate two more advanced sharpness-aware optimization methods: PGN that combines normal gradients and SAM gradients (Zhao et al., 2022), and Ramdom SAM (RSAM) which adds random perturbations before finding the adversarial directions (Liu et al., 2022b). We let PGN and RSAM run the same amount of time as TSAM on the same computing platform. On all the datasets, we tuned $t$ values via grid search from $\{0, 1, 5, 20, 100\}$.

Our results are shown in Table 1 below. The performance for all $t$'s on three image datasets and different model architectures are reported in Section 5.3. For the GLUE benchmark, we report the standard metrics for each dataset in GLUE. TSAM consistently achieves higher test performance than ERM and variants of SAM. We provide corresponding convergence plots of ERM, vanilla SAM, and TSAM in Appendix C.

## 5.2 FLATNESS OF TSAM SOLUTIONS

In this part, we take a more detailed look into the properties of TSAM solutions compared with the ones of ERM and SAM on the CIFAR100 dataset trained by ResNet18 from scratch. In Figure 2, we plot the loss mean and variance over the neighborhood areas around local minima obtained by different objectives, i.e., $\mathbb{E}[L(\theta^* + \epsilon)]$ and $\text{var}[L(\theta^* + \epsilon)]$, where $\epsilon \sim \mathcal{N}(0, \delta^2)$, and $\theta^*$ denotes the different solutions of any objective (with a slight abuse of notation). These measurements have appeared in prior works named average-loss sharpness (Chen et al., 2021; Wen et al., 2024), and are consistent with our sharpness definition (Definition 3) mentioned before. In Figure 2, for all $\delta$ values, we see that TSAM consistently result in flatter local minima than ERM and SAM measured by both the mean and variance of losses around the minima. In addition, we evaluate sharpness following

Table 1: TSAM achieves higher test performance relative to ERM and different variants of SAM across image datasets and the GLUE benchmark with both CNNs and transformer-based models. TSAM (or SAM) is particularly suitable for applications with distribution shifts (DTD and noisy CIFAR100 datasets), which is also consistent with observations in prior works (Baek et al., 2024; Foret et al., 2020). Convergence plots are shown in Figure 7 in the appendix. TSAM also results in lower test loss, discussed in detail in the next section. Additional Tiny Imagenet results and results with standard deviation are given in Appendix C.

| datasets | models | ERM | SAM | ESAM1 | ESAM2 | PGN | RSAM | TSAM |
|---|---|---|---|---|---|---|---|---|
| CIFAR100 | ResNet18 | 0.7139 | 0.7652 | 0.7740 | 0.7752 | 0.7745 | 0.7735 | **0.7778** |
| | WideResNet16-8 | 0.7322 | 0.7844 | 0.8022 | 0.7903 | 0.7858 | 0.7902 | **0.8085** |
| DTD | ViT finetuning | 0.6638 | 0.6787 | 0.6818 | 0.6835 | 0.6776 | 0.6835 | **0.6882** |
| | WideResNet16-8 | 0.1697 | 0.1745 | 0.1767 | 0.1771 | 0.1823 | 0.1766 | **0.1863** |
| Noisy CIFAR100 | ResNet18 | 0.6101 | 0.6900 | 0.6920 | 0.6727 | 0.6568 | 0.6931 | **0.6998** |
| | WideResNet16-8 | 0.5703 | 0.6802 | 0.6979 | 0.6683 | 0.6402 | 0.6593 | **0.7026** |

| objectives | CoLA | WNLI | SST-2 | MNLI | QNLI | RTE | MRPC | QQP | STSB | **AVG** |
|---|---|---|---|---|---|---|---|---|---|---|
| ERM | 0.52/0.8034 | 0.5493 | 0.9048 | 0.796 | **0.8772** | **0.6065** | 0.8382 | 0.8632 | 0.866/0.863 | 0.7715 |
| SAM | 0.52/0.8048 | **0.5634** | **0.9174** | **0.811** | 0.8642 | 0.5884 | 0.8529 | 0.8771 | **0.870**/0.865 | 0.7756 |
| TSAM1 | 0.52/0.8044 | **0.5634** | 0.9163 | **0.811** | 0.8618 | 0.5902 | 0.8531 | 0.8769 | **0.871**/0.867 | 0.7759 |
| TSAM2 | 0.52/0.8053 | **0.5634** | 0.9163 | **0.812** | 0.8629 | 0.5925 | **0.8580** | 0.8747 | 0.868/0.865 | 0.7762 |
| TSAM | 0.52/**0.8081** | **0.5634** | **0.9186** | **0.811** | **0.8781** | **0.6065** | 0.8505 | **0.8877** | **0.871**/0.866 | **0.7801** |

other common notions by investigating the top-5 eigen values of Hessian (e.g., Foret et al., 2020). Under the same model setup, the top-5 eigenvalues are {342.11, 304.72, 260.71, 252.92, 210.88} for ERM, {232.60, 198.35, 182.61, 153.74, 145.76} for SAM, and {140.91, 113.38, 105.90, 92.94, 89.55} for TSAM ($t$=20). We see that TSAM achieves the smallest max eigenvalues among the three.

We further report the training and test performance of best-tuned ERM, SAM, and TSAM in Table 3 in the appendix. We show that ERM solutions have lower training losses but higher test losses than SAM and TSAM when evaluated on the average test performance (i.e., the 'ERM' column in the right table). This is due to the fact that ERM does not generalize as well as SAM or TSAM, and there exist bad sharp local minima around ERM solutions. On the other hand, while TSAM's average training loss is the highest (which is expected because it does not directly optimize over ERM), the test losses of TSAM

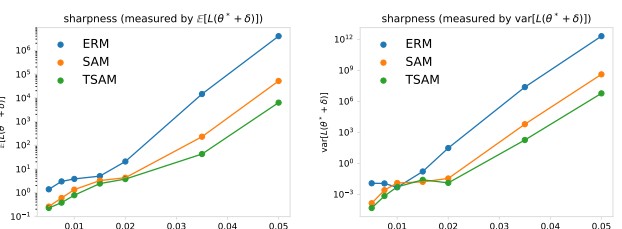

Figure 2: Sharpness of the solutions found by ERM, SAM, and TSAM on CIFAR100 with ResNet18. We empirically measure sharpness by both $\mathbb{E}[L(\theta^* + \epsilon)]$ and $\mathrm{var}[L(\theta^* + \epsilon)]$ where $\epsilon \sim \mathcal{N}(0, \delta^2)$. $\theta^*$ denotes different optimal model parameters obtained from the three objectives. These metrics (especially variance) are also consistent with Definition 3 with $t = 0$. We see that TSAM solutions have a flatter neighborhood compared with the other two.

evaluated by both the average-case performance and worst-case performance are lower than the other two baselines. While we show better generalization of TSAM empirically, rigorous understandings between generalization and flatness is still an open area of research.

## 5.3 SENSITIVITY TO HYPERPARAMETERS

**Effects of the Tilting Hyperparameter $t$.** One critical hyperparameter in TSAM is $t$. When $t = 0$, TSAM objective reduces to the average-case perturbed objective. When $t \to \infty$, the TSAM objective (Eq. (3)) recovers SAM (Eq. 2). But the TSAM algorithm (Algorithm 2) do not exactly recover SAM's alternating updating approximation when $t \to \infty$. See Section 4 for a detailed discussion. Here, we report the test accuracies as the training proceeds under multiple values of $t$'s for all the three tasks. Results are plotted in Figure 3. We see that there are a range of $t$'s that result in faster convergence or higher accuracies than SAM. There also exists an optimal $t$ that leads to the best test performance. This is consistent with our previous generalization bound (Section 3.3). Though Theorem 1 captures the benefits of both SAM and TSAM, we note that the final empirical

performance does not only depend on the properties of the objectives. But rather, it also relies on the choice of approximation algorithms. Results in Theorem 1 assume that the objectives are optimized perfectly, which is infeasible in high-dimensional settings.

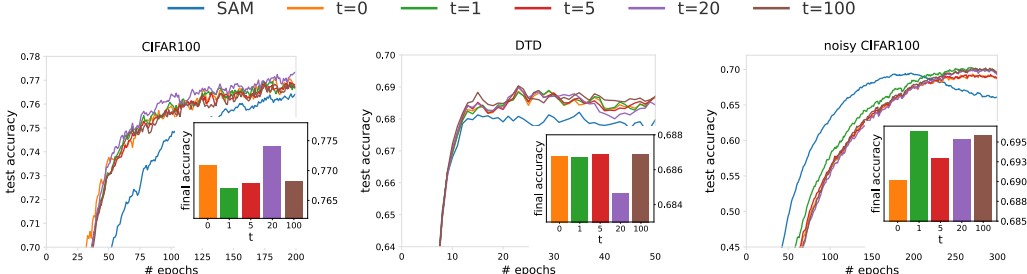

Figure 3: Test accuracies of SAM and TSAM for various values of $t$ when the number of sampled $\epsilon$'s is 3 for each dataset. We select the best SAM and TSAM runs based on the final accuracies on validation data. The results suggest that (1) there are multiple $t$ values that give superior performance than SAM; and (2) we typically need to manually tune a best $t$ via grid search. The empirical performance under different values of $t$ relies on tradeoffs between optimization efficiency (Section 3.1, Section 4), flatness (Section 3.2), and generalization (Section 3.3), and it is difficult to determine an optimal $t$ prior to training. Note that in the label noise regime (left subfigure), one might think that SAM performance could be further improved via a smaller learning rate or early stopping; however, we observe that SAM with a smaller learning rate does not give better accuracy. With early stopping, SAM accuracy is 0.6918, which is still 0.4% lower than that of TSAM without early stopping.

**Effects of Scheduling $t$.** We report all results on TSAM where we fix $t$ values during optimization. Here, we empirically study the effects of scheduling $t$. Increasing $t$ from 0 to a fixed value effectively switches from weighting local minima uniformly to rewighting them based on the loss values, and vice versa. We experiment with two options: linearly decreasing $t$ and linearly increasing $t$ on the noisy CIFAR100 dataset trained by ResNet18. The convergence curves are shown in Figure 4. We see that using a fixed $t$ throughout training does not have significant difference from scheduling $t$. Hence, we stick to fixed $t$'s for our TSAM experiments.

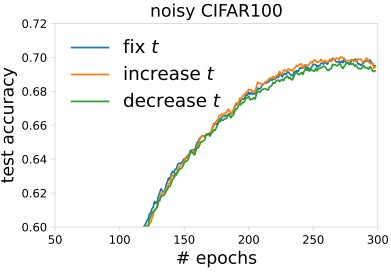

Table 2: Test accuracies of SAM and TSAM with different number of sampled $\epsilon$'s (denoted as $s$; see Algorithm 2). Both $s = 3$ and $s = 5$ leads to improvements over SAM. Empirically, we do not need many samples from the tilted distribution. TSAM performance reported in Table 1 are based on $s = 5, 3, 5$ for the three tasks.

Figure 4: Linearly decreasing or increasing the tilting hyperparameter $t$ with the epochs does not differ from the results of a fixed $t$.

|  | CIFAR100 | DTD | noisy CIFAR100 |
|---|---|---|---|
| SAM | 0.7652 | 0.6787 | 0.6900 |
| TSAM, $s=3$ | 0.7740 | **0.6882** | 0.6955 |
| TSAM, $s=5$ | **0.7778** | 0.6870 | **0.6998** |

**Effects of the Number of $\epsilon$'s.** One may wonder whether we need to sample a large number of perturbations for the algorithm to be effective. In Table 2, we show that we usually only need $s = 3$ or 5 number of $\epsilon$'s to achieve significant improvements relative to SAM.

## 6 CONCLUSION

In this work, we have proposed a tilted sharpness-aware minimization (TSAM) objective, which leverages exponential tilting (parameterized by $t$) to reweight potentially many local minima in the neighborhoods, as opposed to the worst-case minima SAM targets at. We have proved TSAM is a more smooth problem relative to SAM with a bounded $t$, and that TSAM explicitly encourages flatter solutions as $t$ increases for a class of problems including generalized linear models. We have proposed a practical algorithm motivated by HMC to sample from the tilted distribution $e^{tL(\theta+\epsilon)}$. Through experiments on different models and datasets including label-noise settings, we have demonstrated that TSAM consistently outperforms SAM and its variants on both image and text datasets.

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

# A  ADDITIONAL TOY PROBLEMS

In Figure 1 in Section 1, we present a specific toy problem where TSAM arrives at more flat solutions as $t$ increases. Though the TSAM objective will recover SAM when $t \to \infty$, we note that TSAM can be easier to solve due to smoothness. To illustrate this, we create another toy problem in Figure 5 and 6 below. We see that SAM always leads to a non-smooth optimization problem for $\rho > 0$.

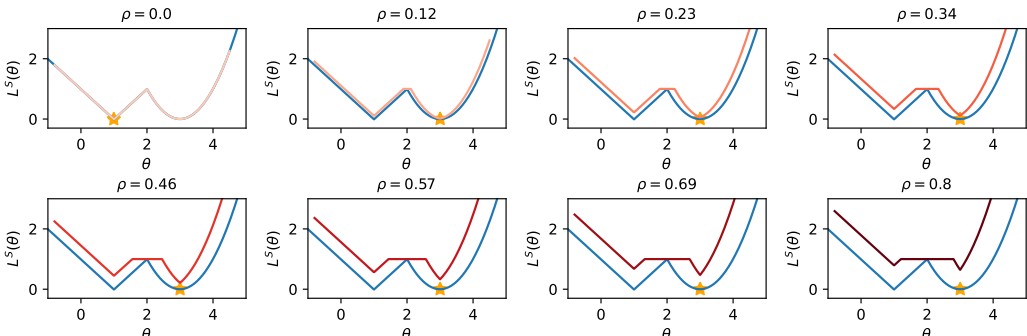

Figure 5: **SAM losses as $\rho$ increases.** The original loss function (shown in the blue lines across all figures) is a one-dimensional problem $L(\theta) = |\theta - 1| - 0.01$ *if* $\theta \leqslant 2$, and $L(\theta) = (\theta - 3)^2$ *otherwise*. Note that $\theta = 3$ is a more flat solution than $\theta = 1$, though $L(1) < L(3)$. The SAM objective is $\min_\theta \max_{|\epsilon| \leqslant \rho} L(\theta + \epsilon)$, shown in the red lines, where the values of $\rho$'s increase from 0 to 0.8. When $\rho = 0$, the objective reduces to ERM. For $\rho > 0$, the SAM objectives (red lines) are non-smooth, and the global minima (marked in orange) are achieved at a flat region in $L(\cdot)$. The SAM objective visualization holds *regardless of the usage of any existing SAM algorithms or implementation.*

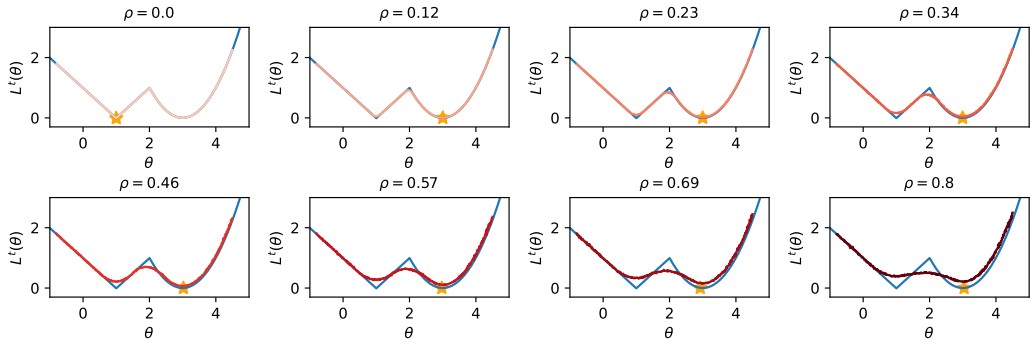

Figure 6: **TSAM ($t$=0.01) losses as $\rho$ increases.** The TSAM objective (red lines) is $\frac{1}{t} \log \left( \mathbb{E}_{\mu(\epsilon)} \left[ e^{tL(\theta + \epsilon)} \right] \right)$, where $\mu(\epsilon) := \mathcal{U}(|\epsilon| \leqslant \rho)$ defines a uniform distribution of $\epsilon$'s constrained in a ball with radius $\rho$. The larger $\rho$ is, the darker the redness becomes. TSAM with a small $t$ is able to find flat solutions.

## B  COMPLETE PROOFS

### B.1  PROOFS FOR SECTION 3.1

**Proof for the Case of** $t \to 0$, $L^t(\theta + \epsilon) \to \mathbb{E}[L(\theta + \epsilon)]$.  Note that if $L(\cdot)$ is continuously differentiable, then $e^{tL(\theta + \epsilon)}$ is continuous w.r.t. $\epsilon \in \mathbb{R}^d$. It is also continuous w.r.t. $t \in \mathbb{R}$. When $t \to 0$,

$$\lim_{t \to 0} L^t(\theta) = \lim_{t \to 0} \frac{1}{t} \log \left( \int e^{tL(\theta + \epsilon)} d\mu(\epsilon) \right) \tag{12}$$

$$= \frac{1}{\int e^{tL(\theta + \epsilon)} d\mu(\epsilon)} \int e^{tL(\theta + \epsilon)} L(\theta + \epsilon) d\mu(\epsilon) \tag{13}$$

$$= \int L(\theta + \epsilon) d\mu(\epsilon) \tag{14}$$

$$= \mathbb{E}[L(\theta + \epsilon)]. \tag{15}$$

**Proof for Lipschitzness.**  First observe that if $L(\theta)$ is $p$-Lipschitz with respect to $\theta$, then $L^t(\theta)$ is $p$-Lipschitz with respect to $\theta$. This follows from

$$\left| L^t(\theta_1) - L^t(\theta_2) \right| = \left| \frac{1}{t} \log \mathbb{E} \left[ e^{tL(\theta_1 + \epsilon)} \right] - \frac{1}{t} \log \mathbb{E} \left[ e^{tL(\theta_2 + \epsilon)} \right] \right| \tag{16}$$

$$= \left| \frac{1}{t} \log \frac{\mathbb{E} \left[ e^{tL(\theta_1 + \epsilon)} \right]}{\mathbb{E} \left[ e^{tL(\theta_2 + \epsilon)} \right]} \right| \tag{17}$$

$$\leqslant \left| \frac{1}{t} \log e^{tp\|\theta_1 - \theta_2\|} \frac{\mathbb{E} \left[ e^{tL(\theta_2 + \epsilon)} \right]}{\mathbb{E} \left[ e^{tL(\theta_2 + \epsilon)} \right]} \right| \tag{18}$$

$$= p\|\theta_1 - \theta_2\|. \tag{19}$$

**Proof for Strong Convexity.**  Assume $L$ is continuously differentiable. If $L$ is $\mu$-strongly convex, then $L^t$ is also $\mu$-strongly convex. This is because of the Hessian in Eq. (20), which can be written as

$$\nabla^2 L^t(\theta) = t \cdot M + \mathbb{E} \left[ \frac{e^{tL(\theta + \epsilon)}}{\mathbb{E}[e^{tL(\theta + \epsilon)}]} \nabla^2 L(\theta + \epsilon) \right], \tag{20}$$

where $M$ is a positive semi-definite matrix. We note that due to the $\mu$-strong convexity of $L$, the second term satisfies $\mathbb{E} \left[ \frac{e^{tL(\theta + \epsilon)}}{\mathbb{E}[e^{tL(\theta + \epsilon)}]} \nabla^2 L(\theta + \epsilon) \right] \succcurlyeq \mu \mathbf{I}$. Hence, $\nabla^2 L^t(\theta) \succcurlyeq \mu \mathbf{I}$.

**Proof for Smoothness.**  From Eq. (20), we know that

$$\frac{1}{t} \nabla^2 L^t(\theta) = M + \frac{1}{t} \mathbb{E} \left[ \frac{e^{tL(\theta + \epsilon)}}{\mathbb{E}[e^{tL(\theta + \epsilon)}]} \nabla^2 L(\theta + \epsilon) \right]. \tag{21}$$

As $\mathbb{E} \left[ \frac{e^{tL(\theta + \epsilon)}}{\mathbb{E}[e^{tL(\theta + \epsilon)}]} \right] = 1$, and the max eigenvalue $\lambda_{\max}(\nabla^2 L(\theta + \epsilon)) \leqslant \beta$, we have

$$0 < \min_{t \to \infty} \frac{1}{t} \lambda_{\max}(\nabla^2 L(\theta + \epsilon)) < +\infty. \tag{22}$$

### B.2  PROOFS FOR SECTION 3.2

In the following, we use $\mathbb{E}$ to denote $\mathbb{E}_\epsilon$. Define $g(t)$ as

$$g(t) := L^t(\theta_1) - L^t(\theta_2) \tag{23}$$

$$= \frac{1}{t} \log \left( \int e^{tL(\theta_1 + \epsilon)} d\mu(\epsilon) \right) - \frac{1}{t} \log \left( \int e^{tL(\theta_2 + \epsilon)} d\mu(\epsilon) \right). \tag{24}$$

Assume $l$ has the specific form of

$$l(x_i; \theta) = A(\theta) - \theta^\top T(x_i), \tag{25}$$

$$L(\theta) = A(\theta) - \theta^\top \left(\frac{1}{n}\sum_{i=1}^{n} T(x_i)\right) := A(\theta) - \theta^\top T(x). \tag{26}$$

Under this form, we have that

$$L^t(\theta) = \frac{1}{t}\log\left(\int e^{t(A(\theta+\epsilon)-(\theta+\epsilon)^\top T(x))} p(\epsilon)d\epsilon\right) \tag{27}$$

$$= \frac{1}{t}\log\left(e^{-t\theta^\top T(x)}\int e^{t(A(\theta+\epsilon)-\epsilon^\top T(x))} p(\epsilon)d\epsilon\right) \tag{28}$$

$$= -\theta^\top T(x) + \frac{1}{t}\log\left(\int e^{t(A(\theta+\epsilon)-\epsilon^\top T(x))} p(\epsilon)d\epsilon\right) \tag{29}$$

Define

$$\Gamma^t(\theta) := \log\left(\mathbb{E}\left[e^{tA(\theta+\epsilon)-t\epsilon^\top T(x)}\right]\right). \tag{30}$$

Then we have

$$L^t(\theta) = \frac{1}{t}\log\left(\mathbb{E}\left[e^{t\left(A(\theta+\epsilon)-(\theta+\epsilon)^\top T(x)\right)}\right]\right) = -\theta^\top T(x) + \frac{1}{t}\Gamma^t(\theta). \tag{31}$$

Define

$$n^t(\theta) := e^{tA(\theta+\epsilon)-t\epsilon^\top T(x)}, \tag{32}$$

$$h^t(\theta) := \frac{\mathbb{E}\left[n^t(\theta)(A(\theta+\epsilon)-\epsilon^\top T(x)\right]}{\mathbb{E}\left[n^t(\theta)\right]}, \tag{33}$$

$$m^t(\theta) := -\frac{1}{t^2}\Gamma^t(\theta) + \frac{1}{t}h^t(\theta). \tag{34}$$

We have that

$$\frac{\partial n^t(\theta)}{\partial t} = n^t(\theta)(A(\theta+\epsilon)-\epsilon^\top T(x)). \tag{35}$$

We know

$$\frac{\partial \Gamma^t(\theta)}{\partial t} = \frac{\mathbb{E}\left[e^{tA(\theta+\epsilon)-t\epsilon^\top T(x)}(A(\theta+\epsilon)-\epsilon^\top T(x))\right]}{\mathbb{E}\left[e^{tA(\theta+\epsilon)-t\epsilon^\top T(x)}\right]} = h^t(\theta), \tag{36}$$

$$\frac{\partial L^t(\theta)}{\partial t} = -\frac{1}{t^2}\Gamma^t(\theta) + \frac{1}{t}\frac{\partial \Gamma^t(\theta)}{\partial t} = -\frac{1}{t^2}\Gamma^t(\theta) + \frac{1}{t}h^t(\theta) = m^t(\theta). \tag{37}$$

## B.3 $L^t(\theta)$ IS MONOTONICALLY NON-INCREASING AS $t$

We would like to prove the sign of $\frac{\partial L^t(\theta)}{\partial t}$, or $m^t(\theta)$, is non-negative. The sign of $m^t(\theta)$ is the same as the sign of $t^2 m^t(\theta)$. We have

$$t^2 m^t(\theta) = -\Gamma^t(\theta) + th^t(\theta), \tag{38}$$

$$\frac{\partial(t^2 m^t(\theta))}{\partial t} = -h^t(\theta) + h^t(\theta) + t\frac{\partial h^t(\theta)}{\partial t} = t\frac{\partial h^t(\theta)}{\partial t}, \tag{39}$$

and

$$\frac{\partial h^t(\theta)}{\partial t} = \frac{\mathbb{E}[n^t(\theta)(A(\theta+\epsilon)-\epsilon^\top T(x))^2]\mathbb{E}[n^t(\theta)]}{(\mathbb{E}\left[n^t(\theta)\right])^2} \tag{40}$$

$$- \frac{\mathbb{E}[n^t(\theta)(A(\theta+\epsilon)-\epsilon^\top T(x))]\mathbb{E}[n^t(\theta)(A(\theta+\epsilon)-\epsilon^\top T(x))]}{(\mathbb{E}\left[n^t(\theta)\right])^2}. \tag{41}$$

Let random variables $X = \sqrt{n^t(\theta)}(A(\theta + \epsilon) - \epsilon^\top T(x))$, and $Y = \sqrt{n^t(\theta)}$. Following the fact $\mathbb{E}[X^2]\mathbb{E}[Y^2] - (\mathbb{E}[XY])^2 \geqslant 0$ gives

$$\frac{\partial h^t(\theta)}{\partial t} = \frac{\mathbb{E}[X^2]\mathbb{E}[Y^2] - \mathbb{E}[XY]\mathbb{E}[XY]}{(\mathbb{E}\left[n^t(\theta)\right])^2} \geqslant 0. \tag{42}$$

Therefore

$$\frac{\partial(t^2 m^t(\theta))}{\partial t} \geqslant 0. \tag{43}$$

We note that $\lim_{t \to 0} t^2 m^t(\theta) = \lim_{t \to 0} t^2 - \Gamma^t(\theta) + th^t(\theta) = 0$. Hence $t^2 m^t(\theta) \geqslant 0$. Therefore, $m^t(\theta) \geqslant 0$. we have shown that the tilted SAM loss $L^t(\theta)$ is monotonically non-decreasing as the increase of $t$, for any $\theta$.

### B.4 $t$-SAM PREFERS FLATTER MODELS AS $t$ INCREASES

Next, we examine $g^t(\theta_1, \theta_2) := L^t(\theta_1) - L^t(\theta_2)$.

$$\frac{\partial g^t(\theta_1, \theta_2)}{\partial t} = \frac{\partial L^t(\theta_1)}{\partial t} - \frac{\partial L^t(\theta_2)}{\partial t} \tag{44}$$

$$= -\frac{1}{t^2}\Gamma^t(\theta_1) + \frac{1}{t}h^t(\theta_1) + \frac{1}{t^2}\Gamma^t(\theta_2) - \frac{1}{t}h^t(\theta_2) \tag{45}$$

$$= m^t(\theta_1) - m^t(\theta_2) \tag{46}$$

Similarly,

$$\frac{\partial(t^2 m^t(\theta_1))}{\partial t} - \frac{\partial(t^2 m^t(\theta_2))}{\partial t} = t\frac{\partial h^t(\theta_1)}{\partial t} - t\frac{\partial h^t(\theta_2)}{\partial t}. \tag{47}$$

For $t \geqslant 0$,

$$\text{sign}\left(\frac{\partial h^t(\theta_1)}{\partial t} - \frac{\partial h^t(\theta_2)}{\partial t}\right) = \text{sign}\left(\frac{\partial(t^2 m^t(\theta_1))}{\partial t} - \frac{\partial(t^2 m^t(\theta_2))}{\partial t}\right), \tag{48}$$

$$\text{sign}\left(m^t(\theta_1) - m^t(\theta_2)\right) = \text{sign}\left(t^2 m^t(\theta_1) - t^2 m^t(\theta_2)\right). \tag{49}$$

Let the random variable $L_1$ denote $A(\theta_1 + \epsilon) - \epsilon^\top T(x)$, and random variable $L_2$ denote $A(\theta_2 + \epsilon) - \epsilon^\top T(x)$. Then

$$\frac{\partial h^t(\theta_1)}{\partial t} = \frac{\mathbb{E}\left[e^{tL_1}L_1^2\right]\mathbb{E}\left[e^{tL_1}\right] - \left(\mathbb{E}\left[e^{tL_1}L_1\right]\right)^2}{\left(\mathbb{E}\left[e^{tL_1}\right]\right)^2} = \frac{\mathbb{E}\left[e^{tL_1}L_1^2\right]}{\mathbb{E}\left[e^{tL_1}\right]} - \left(\frac{\mathbb{E}[e^{tL_1}L_1]}{\mathbb{E}[e^{tL_1}]}\right)^2, \tag{50}$$

$$\frac{\partial h^t(\theta_1)}{\partial t} - \frac{\partial h^t(\theta_2)}{\partial t} = \frac{\mathbb{E}\left[e^{tL_1}L_1^2\right]}{\mathbb{E}\left[e^{tL_1}\right]} - \left(\frac{\mathbb{E}[e^{tL_1}L_1]}{\mathbb{E}[e^{tL_1}]}\right)^2 - \left(\frac{\mathbb{E}\left[e^{tL_2}L_2^2\right]}{\mathbb{E}\left[e^{tL_2}\right]} - \left(\frac{\mathbb{E}[e^{tL_2}L_2]}{\mathbb{E}[e^{tL_2}]}\right)^2\right). \tag{51}$$

Given random variables $L_1$ and $L_2$, the *exponentially reweighted* losses can be defined as $e^{tL_1}L_1$ and $e^{tL_2}L_2$. The *t-weighted* second moment is $\frac{\mathbb{E}\left[e^{tL_1}L_1^2\right]}{\mathbb{E}[e^{tL_1}]}$, and the *t-weighted* mean is $\frac{\mathbb{E}[e^{tL_1}L_1]}{\mathbb{E}[e^{tL_1}]}$. Hence, $\frac{\mathbb{E}\left[e^{tL_1}L_1^2\right]}{\mathbb{E}[e^{tL_1}]} - \left(\frac{\mathbb{E}[e^{tL_1}L_1]}{\mathbb{E}[e^{tL_1}]}\right)^2$ can be viewed as *t-weighted* variance. As $\theta_1$ is $t$-sharper than $\theta_2$, we have $\frac{\partial h^t(\theta_1)}{\partial t} - \frac{\partial h^t(\theta_2)}{\partial t} \geqslant 0$. Therefore $t^2 m^t(\theta_1) - t^2 m^t(\theta_2)$ is non-decreasing as $t$ increases. It takes value of 0 when $t = 0$, which implies that $t^2 m^t(\theta_1) - t^2 m^t(\theta_2) = \frac{\partial g^t(\theta_1, \theta_2)}{\partial t} \geqslant 0$.

**Proof for the Discussions on $t \to 0$.** Recall that exp and log functions can be expanded as

$$\exp(x) = 1 + \sum_{k=1}^{\infty} \frac{x^k}{k!} \approx 1 + x + \frac{1}{2}x^2, \tag{52}$$

$$\log(x + 1) = \sum_{k=1}^{\infty} (-1)^{k-1}\frac{x^k}{k!} \approx x - \frac{x^2}{2} + \frac{x^3}{6}. \tag{53}$$

For very small $t \leqslant 1$,

$$\frac{1}{t} \log \left( \mathbb{E} \left[ e^{tL(\theta + \epsilon)} \right] \right) \tag{54}$$

$$\approx \frac{1}{t} \log \left( \mathbb{E} \left[ 1 + tL(\theta + \epsilon) + \frac{t^2}{2} L^2(\theta + \epsilon) \right] \right) \tag{55}$$

$$\approx \frac{1}{t} \left( \mathbb{E} \left[ tL(\theta + \epsilon) + \frac{t^2}{2} L^2(\theta + \epsilon) \right] - \frac{1}{2} \mathbb{E}^2 \left[ tL(\theta + \epsilon) + \frac{t^2}{2} L^2(\theta + \epsilon) \right] \right) \tag{56}$$

$$= \frac{1}{t} \left( t\mathbb{E}[L(\theta + \epsilon)] + \frac{t^2}{2} \mathbb{E} \left[ L^2(\theta + \epsilon) \right] - \frac{t^2}{2} \mathbb{E}^2[L(\theta + \epsilon)] + O(t^3) + O(t^4) \right) \tag{57}$$

$$= \mathbb{E}[L(\theta + \epsilon)] + \frac{t}{2} \left( \mathbb{E} \left[ L^2(\theta + \epsilon) \right] - \mathbb{E}^2 \left[ L(\theta + \epsilon) \right] \right) + O(t^2) \tag{58}$$

$$\approx \mathbb{E}[L(\theta + \epsilon)] + \frac{t}{2} \mathrm{var} \left( L(\theta + \epsilon) \right) \tag{59}$$

Hence, our proposed objective can be viewed as optimizing for the mean plus variance of the losses in the neighborhood regions when $t$ is very close to 0. When $t = 0$, it reduces to only optimizing for $\mathbb{E}[L(\theta + \epsilon)]$ for $\epsilon \in \mu(\epsilon)$. For any $\theta_1$ and $\theta_2$ such that $\theta_1$ is sharper than $\theta_2$,

$$g(t) \approx \mathbb{E}[L(\theta_1 + \epsilon)] + \frac{t}{2} \mathrm{var} \left( L(\theta_1 + \epsilon) \right) - \mathbb{E}[L(\theta_2 + \epsilon)] - \frac{t}{2} \mathrm{var} \left( L(\theta_2 + \epsilon) \right), \tag{60}$$

$$g'(t) \approx \frac{1}{2} \left( \mathrm{var} \left( L(\theta_1 + \epsilon) \right) - \mathrm{var} \left( L(\theta_2 + \epsilon) \right) \right) \tag{61}$$

Sharpness is defined as standard variance when $t \to 0$ (Definition 3), and we have that $g'(t) \geqslant 0$.

### B.5 PROOF FOR THEOREM 2

We first state some useful lemmas.

**Lemma 3** (Aminian et al. (2024)). *Let $X$ be a random variable. Suppose $0 < a < X < b < +\infty$, we have*

$$\frac{\mathrm{var}(X)}{2b^2} \leqslant \log(\mathbb{E}[X]) - \mathbb{E}[\log(X)] \leqslant \frac{\mathrm{var}(X)}{2a^2}. \tag{62}$$

The Lemma directly follows from existing results in Aminian et al. (2024). For completeness, we include the proof here.

*Proof.* As $\frac{d^2}{dx^2} \left( \log(x) + \beta x^2 \right) = \frac{-1}{x^2} + 2\beta$, the function $\log(x) + \beta x^2$ is concave for $\beta = \frac{1}{2b^2}$ and convex for $\beta = \frac{1}{2a^2}$. Hence, by Jensen's inequality,

$$\mathbb{E}[\log(X)] = \mathbb{E} \left[ \log(X) + \frac{X^2}{2b^2} - \frac{X^2}{2b^2} \right] \leqslant \log(\mathbb{E}[X]) + \frac{1}{2b^2} \mathbb{E}[X]^2 - \frac{1}{2b^2} \mathbb{E}[X^2] \tag{63}$$

$$= \log(\mathbb{E}[X]) - \frac{1}{2b^2} \mathrm{var}(X), \tag{64}$$

which completes the proof of the lower bound. A similar approach can be applied to derive the upper bound. $\qquad\square$

**Proof for Theorem 2.** We can now proceed with the detailed proof below.

*Proof.* Examine the following decomposition of the generalization error

$$\mathbb{E}_Z[l(\theta, Z)] - \frac{1}{t} \log \mathbb{E}_\epsilon[e^{tL(\theta + \epsilon)}] \tag{65}$$

$$= \mathbb{E}_Z[l(\theta, Z)] - \mathbb{E}_\epsilon[L(\theta + \epsilon)] + \mathbb{E}_\epsilon[L(\theta + \epsilon)] - \frac{1}{t} \log \mathbb{E}_\epsilon[e^{tL(\theta + \epsilon)}] \tag{66}$$

Based on Lemma 3, let $X$ be $e^{tL(\theta+\epsilon)}$ and $1 \leqslant e^{tL(\theta+\epsilon)} \leqslant e^{tM}$ (assuming positive and bounded losses and non-negative $t$), we have that

$$\frac{\text{var}(e^{tL(\theta+\epsilon)})}{2te^{2tM}} \leqslant \frac{1}{t}\log(\mathbb{E}_\epsilon[e^{tL(\theta+\epsilon)}]) - \mathbb{E}[L(\theta+\epsilon)] \leqslant \frac{\text{var}(e^{tL(\theta+\epsilon)})}{2t}, \tag{67}$$

and

$$-\frac{\text{var}(e^{tL(\theta+\epsilon)})}{2te^{2tM}} \geqslant \mathbb{E}_\epsilon[L(\theta+\epsilon)] - \frac{1}{t}\log(\mathbb{E}_\epsilon[e^{tL(\theta+\epsilon)}]) \geqslant -\frac{\text{var}(e^{tL(\theta+\epsilon)})}{2t}. \tag{68}$$

For the term $\mathbb{E}_Z[l(\theta,Z)] - \mathbb{E}_\epsilon[L(\theta+\epsilon)]$, we further decompose it into

$$\mathbb{E}_Z[l(\theta,Z)] - \mathbb{E}_\epsilon[L(\theta+\epsilon)] = \mathbb{E}_Z[l(\theta,Z)] - L(\theta) + L(\theta) - \mathbb{E}_\epsilon[L(\theta+\epsilon)]. \tag{69}$$

Recall that $L(\cdot)$ denote the empirical average loss based on $n$ training samples (Eq. (1)), applying Hoeffding Inequality (Boucheron et al., 2013) gives

$$\mathbb{E}_Z[l(\theta,Z)] - L(\theta) \leqslant M\sqrt{\frac{\log(2/\delta)}{2n}}. \tag{70}$$

Combining Eq.(68), (69), and (70), we have the desired bound

$$\mathbb{E}_Z[l(\theta,Z)] - \frac{1}{t}\log\mathbb{E}_\epsilon[e^{tL(\theta+\epsilon)}] \tag{71}$$

$$\leqslant M\sqrt{\frac{\log(2/\delta)}{2n}} - \frac{\text{var}(e^{tL(\theta+\epsilon)})}{2te^{2tM}} + L(\theta) - \mathbb{E}_\epsilon[L(\theta+\epsilon)]. \tag{72}$$

To investigate the impacts of $t$ on the generalization bound, we can leave the last term $L(\theta) - \mathbb{E}_\epsilon[L(\theta+\epsilon)]$ as it is since it is independent of $t$.

Additionally, to bound $L(\theta) - \mathbb{E}_\epsilon[L(\theta+\epsilon)]$ (the gap between empirical average losses and its randomly-smoothed version) we note that it is related to the curvature of $L(\theta)$. If we further assume that the loss is $\mu$-strongly convex, then it holds that

$$L(\theta) - \mathbb{E}[L(\theta+\epsilon)] \leqslant L(\theta) - L(\theta) - \mathbb{E}[\nabla L(\theta)^\top \epsilon] - \frac{\mu}{2}\|\epsilon\|^2 \leqslant -\frac{\mu}{2}\mathbb{E}[\|\epsilon\|^2]. \tag{73}$$

Combining all the above results gives

$$\mathbb{E}_Z[l(\theta,Z)] - \frac{1}{t}\log\mathbb{E}_\epsilon[e^{tL(\theta+\epsilon)}] \leqslant M\sqrt{\frac{\log(2/\delta)}{2n}} - \frac{\text{var}(e^{tL(\theta+\epsilon)})}{2te^{2tM}} - \frac{\mu}{2}\mathbb{E}[\|\epsilon\|^2]. \tag{74}$$

$\square$

# C   Additional Experimental Details

## C.1   Hyperparamter Tuning

We use momentum with a parameter 0.9 for all algorithms. For all the three datasets and all methods, we tune learning rates from $\{0.0001, 0.0003, 0.001, 0.003, 0.01, 0.03, 0.1\}$. We use a fixed batch size of 64, label smoothing of 0.1 (for smooth cross-entropy loss), momentum with a parameter 0.9, and $L_2$ weight decay 0.0005 for all runs. For vanilla SAM, we tune $\rho$ from $\{0.05, 0.1, 0.5, 1, 5, 15\}$ and found that the best values are $0.05, 0.1, 0.1$ for CIFAR100, DTD, noisy CIFAR100, respectively. For SAM variants, we tune $\rho$ parameters in the same way. The PGN baseline (Zhao et al., 2022) introduces another hyperparameter—the coefficients for the linear combination between normal gradients and SAM gradients, and we tune that from $\{0.3, 0.5, 0.7, 0.9\}$. We follow the recommendations of $\lambda$ and $\gamma$ hyperparameters in the original Random SAM paper (Liu et al., 2022b). For TSAM, we set $\delta = \frac{t}{2}$ in Algorithm 1, set $\rho$ to be 20, and $\alpha$ in Algorithm 2 to be 0.995 across all datasets. We tune $t$ from $\{0, 1, 5, 20, 100\}$, and the best $t$'s are 20, 5, 1 for the three image datasets. The number of sampled $\epsilon$'s (the $s$ hyperparameter in Algorithm 1) are chosen from $\{3, 5\}$. We show the effects of $t$ and $s$ in detail in Section 5.3 in the main text.

## C.2   Additional Results

**Convergence Curves.**   In Table 1, we present the final test accuracies of TSAM and the baseline. In Figure 7, we show the convergence of these methods on three datasets. We see that TSAM achieves the fastest convergence, and arrives at a better solution. This is consistent with our argument that TSAM with a bounded t is a more smooth objective than the original SAM formulation (Section 3).

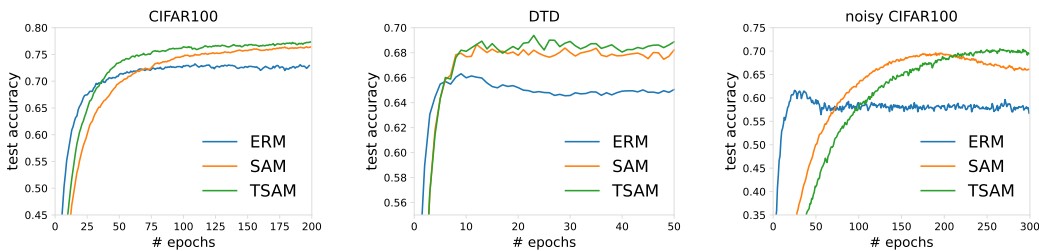

Figure 7: Convergence curves on three image datasets showing test accuracies.

**Training and Test Loss Comparisons under Different Objectives.**   In Table 3, we show that TSAM generates better than ERM and SAM by comparing training and test losses.

Table 3: To further understand TSAM behavior, we report the losses of models trained by {ERM, SAM, TSAM} and evaluated on {ERM, SAM, TSAM} objectives, respectively. The left table shows **training** losses and the right one shows **test** losses. We see that (1) every objective achieves the smallest *training* loss if directly being optimized (diagonal entries, left table). (2) Though SAM and TSAM incurs larger training losses than ERM (last two rows, left table), they lead to smaller test losses (last two rows, right table), i.e., better generalization. (3) Optimizing TSAM results in the smallest test loss across all the three metrics (last row, right table).

| evaluated on / trained on | ERM | SAM | TSAM |
|---|---|---|---|
| | | training loss | |
| ERM | **0.1283** | 1.35 | 3.48 |
| SAM | 0.1489 | **0.22** | 0.60 |
| TSAM | 0.1763 | 0.27 | **0.46** |

| evaluated on / trained on | ERM | SAM | TSAM |
|---|---|---|---|
| | | test loss | |
| ERM | 0.9302 | 2.05 | 3.54 |
| SAM | 0.7414 | 0.91 | 1.34 |
| TSAM | **0.7163** | **0.90** | **1.08** |

**Additional results on Tiny Imagenet.**   Tiny imagenet dataset is a subset of imagenet with 200 classes (Deng et al., 2009). In the table below, we report the results on finetuning on tiny imagenet with ResNet18 starting from a pre-trained model. For TSAM, we sample $s$=3 perturbations. Note

that ESAM1 is simply the vanilla SAM update but running 3x more iterations. We see that TSAM again outperforms the baselines on this dataset.

| metric | ERM | ESAM1 | PGN | TSAM |
|---|---|---|---|---|
| top-1 accuracy | 0.711 (.0009) | 0.7324 (.0007) | 0.7318 (.001) | 0.7355 (.0008) |

Table 4: Results on the tiny imagenet dataset.

**Effects of $N$ in HMC.** Recall that Algorithm 2 follows HMC updates in Eq. (11) by running it for $N = 1$ step. In principle, we can run Eq. (11) for more than 1 step to generate each candidate perturbation $\epsilon$. This require additional gradient evaluations, which can be very expensive. We report results of $N = 2$ and $N = 3$ in Table 5 below.

| configurations | CIFAR100 | DTD |
|---|---|---|
| $N = 1\ (s = 3)$ | 0.7740 | 0.6882 |
| $N = 1\ (s = 5)$ | 0.7778 | 0.6870 |
| $N = 2\ (s = 3)$ | 0.7745 | 0.6883 |
| $N = 3\ (s = 3)$ | 0.7752 | 0.6880 |

Table 5: $N > 1$ requires $N$ times more gradient evaluations to generate a single perturbation $\epsilon$ (Eq. (11), without resulting in significantly better model accuracies.

**Results with Standard Deviation.** We show the standard deviation of our main results on the image datasets in Table 6 below.

| datasets | ERM | SAM | ESAM1 | ESAM2 | PGN | RSAM | TSAM |
|---|---|---|---|---|---|---|---|
| CIFAR100 | 71.39 (.17) | 76.52 (.10) | 77.40 (.14) | 77.52 (.07) | 77.45 (.06) | 77.35 (.13) | **77.78** (.11) |
| DTD | 66.38 (.14) | 67.87 (.15) | 68.18 (.15) | 68.35 (.12) | 67.76 (.12) | 68.35 (.21) | **68.82** (.11) |
| Noisy CIFAR100 | 61.01 (.25) | 69.00 (.12) | 69.20 (.09) | 67.27 (.07) | 65.68 (.14) | 69.31 (.09) | **69.98** (.12) |

Table 6: Full Results (corresponding to Table 1 (top) in the main text) with mean and standard deviation across three runs with different random seeds.

**Runtime Comparisons.** We report the runtime of TSAM and other baselines in Table 7 below.

Table 7: Runtime comparisons (in minutes) averaged over different hyperparameter sets. ERM and SAM run the same numbers of epochs as TSAM. All baselines are explained in Section 5. ESAM1 denotes the baseline of letting SAM run longer until it reaches the same computation budget as TSAM.

| datasets | ERM | SAM | ESAM1 | ESAM2 | PGN | RSAM | TSAM |
|---|---|---|---|---|---|---|---|
| CIFAR100 | 32 | 65 | 325 | 310 | 333 | 320 | 333 |
| DTD | 5 | 6 | 18 | 17 | 15 | 16 | 15 |
| Noisy CIFAR100 | 45 | 92 | 350 | 315 | 302 | 312 | 337 |

### C.3 NAIVE SAMPLING

As discussed in Section 4, one naive approach to estimate $\frac{\mathbb{E}[e^{tL(\theta+\epsilon)}\nabla L(\theta+\epsilon)]}{\mathbb{E}[e^{tL(\theta+\epsilon)}]}$ for $\epsilon$ in uniformly distributed over $\mu(\epsilon)$ is to first uniformly sample $s$ $\epsilon$'s over $\mu(\epsilon)$, and then perform tilted aggregation, as follows:

$$\hat{g} \leftarrow \frac{\sum_{i \in [s]} e^{tL(\theta+\epsilon_i)} \nabla L(\theta + \epsilon_i)}{\sum_{i \in [s]} e^{L(\theta+\epsilon_i)}}, \{\epsilon_i\}_{i \in [s]} \sim \mu(\epsilon). \tag{75}$$

We demonstrate convergence as a function of $s$ on the CIFAR100 dataset in the figure below. We see that as $s$ increases, the performance increases. However, when $s = 10$, which means that we need 10 gradient evaluations per model updates, the accuracy is lower than that of TSAM with the proposed algorithm.

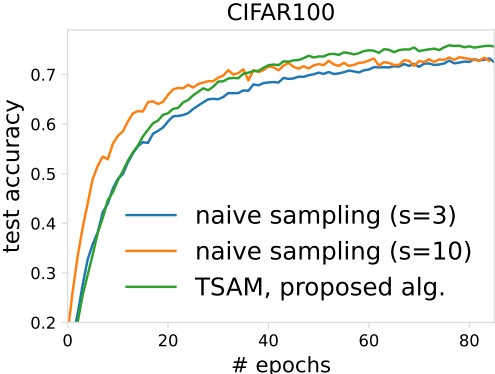

Figure 8: TSAM ($t = 20$) with the proposed Algorithm 1 and 2 and compared with the strategy of uniformly sampling $\epsilon$ and performing tilted aggregation over them.

