# OpenReview forum: "Reweighting Local Mimina with Tilted SAM"
_ICLR.cc/2025/Conference — Submitted to ICLR 2025_

### Official Review · Reviewer_aGF9 · 2024-10-20

**Soundness:** 2
**Presentation:** 2
**Contribution:** 2
**Rating:** 5
**Confidence:** 5

**Summary:**

The paper presents Tilted SAM (TSAM) that generalizes Sharpness-Aware Minimization (SAM) by incorporating exponential tilting. While SAM aims to minimize the worst-case loss in a neighborhood around the model parameters to attain flatter minima, TSAM enhances this approach by emphasizing local solutions that are flatter but allow for larger losses, controlled by a tilt hyperparameter $t$. Compared to SAM, TSAM is less aggressive as it adjusts the perturbation between worst-case scenarios and other states. The paper includes theoretical proof of TSAM's properties, describes an algorithm inspired by Hamiltonian dynamics for solving TSAM, and presents empirical results demonstrating TSAM's superior generalization performance and ability to achieve flatter minima.

**Strengths:**

1. TSAM, a smoother objective for SAM, achieves flatter minima through the use of exponential tilting.

2. The paper defines sharpness in terms of loss variance and provides proof that TSAM can achieve flatter minima.

3. The authors propose a practical algorithm inspired by Hamiltonian Monte Carlo.

4. Experimental results show that TSAM achieves better generalization and attains flatter minima in both image and language classification tasks.

**Weaknesses:**

1. Section 3.2 demonstrates that TSAM prefers a flatter model as $t$ increases. However, the relationship between flatness and generalization is still unclear as the authors mentioned in related works. Although the authors analyze the generalization of TSAM in Section 3.3, it's a bit weak since Remark 1 just claims that TSAM _can_ obtain a smaller linear population error of ERM or SAM.

2. Lack of experiments on large-scale datasets like ImageNet-1K, which is a standard dataset in SAM-related works.

3. In Figure 3, it seems $t$ is a sensitive hyperparameter and it needs to be tuned on different datasets, which adds extra computational cost. In addition, as $t$ increases that should lead to flatter minima as discussed in Section 3.2, but it doesn't always improve the generaliation. How to pick effective $t$ becomes a problem.

4. Why choose $\mathrm{E}[L(\theta^\star + \epsilon)]$ and $\mathrm{var}[L(\theta^\star + \epsilon)]$ representing sharpness in Section 5.2? Hessian spectra, the maximum eigenvalue, and $\lambda_1/\lambda_5$ are the most common methods to describe the sharpness in SAM-related works (e.g. SAM paper and VASSO paper, both of them mentioned in this paper). Therefore, the authors should explain the reason, and that would be better if adding additional results.

5. Computational cost is too much. TSAM requires 3 or 5 times the computational cost of SAM, which makes it impractical. Don't even mention the costs of tuning $t$ and $s$.

6. It's not very clear about experimental settings. For example, what's the noise level of Noisy CIFAR100? Are the results of Table 1 mean values, how many runs of each experiment, and could you provide the standard deviation?

**Questions:**

1. What's the meaning of '(1)' at the end of line 281?

2. What's the role of $\hat{\theta}$ in Algorithm 2? Why not compute $\epsilon$ directly?

3. Is it a typo in line 475? When $t=0$, TSAM reduces to SAM objective or average weight perturbation?

---

> ### Author Response · Authors · 2024-11-20
> **response**
>
> We thank the reviewer for the comments and questions.
>
> **[Generalization]** Theorem 2 implies that TSAM (under some bounded $t$) would result in a smaller upper bound of linear population error compared with ERM, which we think is an interesting result. As we compare the upper bounds of the linear population risk (which is the metric of interest in practical deployment), optimizing for TSAM is optimizing for a smaller upper bound. We provide results (Theorem 1) where TSAM leads to better generalization or flatness; but the relationship between flatness and generalization in deep neural works is still an open problem (Wen et al., 2023).
>
> [Wen et al., 2023] Sharpness Minimization Algorithms Do Not Only Minimize Sharpness To Achieve Better Generalization, NeurIPS 2023.
>
>
> **[Additional experiments]** Thanks for the suggestion of evaluating TSAM on an additional ImageNet dataset. Due to constraints with the setup of our compute resources, we find it’s going to take a long time to run imagenet experiments with proper hyperparameter tuning for all methods. Therefore, we explore the tiny imagenet dataset, which is a subset of imagenet with 200 classes. In the table below, we report the results on finetuning tiny imageNet with ResNet18 starting from a pre-trained model. For TSAM, we sample $s$=3 perturbations. Note that ESAM1 is simply the vanilla SAM update but running $3\times$ more iterations. We see that TSAM again outperforms the baselines on this dataset. This trend is consistent with other results on a variety of image and language benchmarks and models in the paper.
>
> |                 |      ERM   |      ESAM1 | PGN | TSAM |
> |-----------|-------------|-------------|----------|----- |
> |top-1 accuracy | 0.7110 (.0010) | 0.7324 (.0006) | 0.7318 (.002) |0.7354 (.0006) |
>
>
>
> **[How to tune t]** In some cases (DTD and noisy CIFAR100 datasets), there exist a wide range of $t$’s that all result in improvements compared with SAM. But there is an optimal $t$ that leads to the best test performance, which needs manual tuning. In our experiments, we only pick $t$ from a limited set of grids of {0, 1, 5, 20, 100}. Though tuning $t$ requires extra effort, we obtain better generalization on a variety of tasks. Note that for the baselines (e.g., more advanced SAM variants such as PGN or RSAM) that TSAM outperforms, they generally introduce additional one or two new hyperparameters that we also tuned carefully. One goal of this work is to provide some high-level intuitions around the effects of $t$ (Section 3).
>
>
> **[Sharpness definition]** Thanks for the suggestion of evaluating more sharpness definitions. As partially mentioned in Section 5.2, we choose mean and variance because (a) those are intuitive statistics to reason about flatness of local loss surface, (b) they have appeared in prior works called average-perturbation sharpness or average-direction sharpness (e.g., Chen et al., 2022, Wen et al., 2023), and (c) such notions are more aligned with our theoretical analysis in Section 3.2.
>
> Having said that, we agree that it would be better to evaluate on more sharpness measures. We have computed the eigenvalues of the hessian under SAM and TSAM solutions (fixing the same computation budget) of the CIFAR100 dataset trained  ResNet18. Top-5 eigenvalues are {342.11, 304.72, 260.71, 252.92, 210.88} for ERM, {232.60, 198.35, 182.61, 153.74, 145.76} for SAM, and {140.91, 113.38, 105.90,  92.94,  89.55} for TSAM ($t$=20). We see that TSAM achieves the smallest max eigenvalues among the three objectives. We have added more discussions around this and the results to the revision.
>
> [Chen et al., 2022] When Vision Transformers Outperform ResNets without Pre-training or Strong Data Augmentations, ICLR 2022.
> [Wen et al., 2023] Sharpness Minimization Algorithms Do Not Only Minimize Sharpness To Achieve Better Generalization, NeurIPS 2023.

---

> > ### Author Response · Authors · 2024-11-20
> > **response (cont.)**
> >
> > **[Computation cost]** Though TSAM requires more gradient evaluations *per iteration* than the vanilla SAM algorithm, it still outperforms SAM (and SAM variants) when comparing under the same computation budget (e.g., letting baseline SAM algorithms run for more iterations) (Table 1). We report the concrete runtime numbers in the revision in Appendix C.2. We also note that our current implementation can be further improved, as we can sample epsilon’s in parallel to save time.
> >
> > **[Experimental details]** The noise level of CIFAR100 is 20% random label noise, as described in Line 384-385 of the original submission. Results in Table 1 are averaged over three runs with different random seeds that determine model initialization and mini-batch orders. Based on the results in Table 2 where we show that TSAM solutions result in significantly lower test loss than the baselines evaluated by all the three {ERM, SAM, TSAM} losses, we believe our main results are statistically significant. We also added standard deviation results in the revision (Table 5).
> >
> >
> > **[Other questions]** “(1)” is the index for the first observation discussed in the next line. We have fixed it.
> >
> > The updating steps of obtaining $\hat{\theta}$ is an essential step of obtaining perturbation $\epsilon$’s derived from HMC. As discussed in the paper (beginning of Section 4.1 and Figure 8 in the appendix), naively sampling $\epsilon$’s wouldn’t give a good approximation of the tilted gradients.
> >
> > Thanks for catching the typo! When $t$ goes to 0, it reduces to average perturbed loss. We have fixed this.
> >
> > Please let us know if you have additional concerns or questions.

---

> > ### Comment · Reviewer_aGF9 · 2024-11-22
> >
> > Thank you for the authors' response.
> >
> > 1. While the authors emphasize the relationship between flatness and generalization in their reply, I would suggest reducing the focus on TSAM's sharpness analysis (Section 5.2) in the main text to streamline the paper.
> > 2. The experimental results appear somewhat limited. Compared to recent SAM-related works [1-4], training from scratch on ImageNet-1K has become a standard benchmark and is notably absent here. Additionally, many works typically evaluate multiple architectures on each dataset and even explore combinations with other SAM variants.
> > 3. The computational cost of TSAM remains a significant concern. Tuning the parameter $t$ for each model-dataset pair is time-consuming, and TSAM incurs additional computational overhead and memory usage due to multiple perturbations. In contrast, many SAM variants achieve better generalization under the same computational budget. If TSAM requires higher computational resources and cannot be effectively combined with other SAM variants, the practical improvement may appear limited.
> >
> >
> > [1] Jungmin Kwon, Jeongseop Kim, Hyunseo Park, and In Kwon Choi. ASAM: Adaptive sharpness-aware minimization for scale-invariant learning of deep neural networks. In International Conference on Machine Learning (ICML), 2021.
> >
> > [2] Bingcong Li and Georgios Giannakis. Enhancing sharpness-aware optimization through variance suppression. Advances in Neural Information Processing Systems (NeurIPS), 2024.
> >
> > [3] Wanyun Xie, Fabian Latorre, Kimon Antonakopoulos, Thomas Pethick, and Volkan Cevher. Improving SAM requires rethinking its optimization formulation. International Conference on Machine Learning (ICML), 2024.
> >
> > [4] Maximilian Mueller, Tiffany Vlaar, David Rolnick, and Matthias Hein. Normalization layers are all that sharpness-aware minimization needs. Advances in Neural Information Processing Systems (NeurIPS), 2024.

---

> ### Author Response · Authors · 2024-11-25
> **response**
>
> We thank the reviewer for the time and effort spent and letting us know the remaining concerns. We have revised the paper around these comments, and our responses are provided below.
>
> **[Presentation]** Thanks for the suggestion. Part of the confusion may come from the fact that understanding the relations between flatness and generalization is still an active area of research on its own. In our work, we do not claim theoretically flatness *leads to* generalization in our context. Rather, we proved that TSAM helps with both flatness and generalization to the extent described in Theorem 1 and Theorem 2. We have revised Section 5.2 and moved Table 2 to the appendix in the revision.
>
> **[Additional experiments]** We note that while we did not use several model architectures for each dataset, we already have a broad coverage of models (CNNs, vision transformers, and bert models), datasets (noisy and clean versions, three image and one text benchmarks), and evaluation (various baselines, training and test performance, and sharpness of the final solutions) in experiments. We also added tiny imagenet results in Table 4 in the Appendix. Due to computation resource constraints, we will provide complete imagenet results in the next version.
>
> We appreciate the reviewer’s additional comment of evaluating multiple architectures for each dataset. Hence, we perform extra evaluation on the WideResNet model on three datasets for all baselines. Test accuracies are shown in the table below, and have also been added to the revision of the paper (Table 1, Section 5.1).
>
>
> | Dataset (Model)         | ERM    | SAM    | ESAM1  | ESAM2  | PGN    | RSAM   | TSAM (s=3) |
> |--------------------------|--------|--------|--------|--------|--------|--------|------------|
> | CIFAR100 (WideResNet)    | 0.7322 | 0.7844 | 0.8022 | 0.7903 | 0.7858 | 0.7902 | 0.8085     |
> | DTD (WideResNet)         | 0.1697 | 0.1745 | 0.1767 | 0.1771 | 0.1823 | 0.1766 | 0.1863     |
> | CIFAR100 (WideResNet)    | 0.5703 | 0.6802 | 0.6979 | 0.6683 | 0.6402 | 0.6593 | 0.7026     |
>
>
>
> We see that TSAM (number of perturbations $s$=3) still demonstrates superior performance than the baselines under the same computational budget (e.g., TSAM running for 200 epochs and ESAM1 running for 600 epochs).  If we continue to increase the runtime of the baselines, their performance wouldn’t improve.
>
> As discussed in our paper (end of the first paragraph of Section 2), as TSAM is a new sharpness-aware formulation, it’s possible to apply many optimization techniques to solve TSAM, e.g., combining the current TSAM algorithms with other SAM variants. One way is to apply adaptivity or variance reduction on top of the tilted gradients.
>
> We have also included the papers the reviewers mentioned in the paper’s references.
>
> **[Computational cost of TSAM]** We agree that in order to estimate the tilted gradients for TSAM, we generally need more gradient evaluations *per iteration*, as discussed and acknowledged in the paper. However, we have evaluated TSAM and all the baselines with the same computational budget, e.g., by letting baselines run 3$\times$ or 5$\times$ longer than TSAM, or optimizing for the inner max of SAM in a more fine-grained way using more gradient computation (ESAM2). We see that TSAM still outperforms those approaches on a variety of tasks and models. The trend still holds even if we further increase the runtime of ERM (SGD) or SAM. In addition, the implementation of TSAM can be optimized. If we sample the perturbations (which are independent to each other) in parallel, we can incur the same per-iteration runtime as the vanilla SAM algorithm.
>
> We note that other SAM baselines can introduce new hyperparameters as well (e.g., the coefficient parameter in PGN, the noise scaling factor and the ratio between noise and gradients lambda in RSAM). In TSAM, we tune $t$ from a limited set of candidates, and there are multiple $t$’s that can lead to improved performance. In addition, we provide theoretical analysis on the effects of t, which we hope can provide additional intuition. As our responses to Reviewer uH13, one interesting direction is to automatically optimize $t$ as an optimization variable by taking inspirations from previous works (e.g., Eq. (65) in Li et al. (2023)) and connecting the TSAM objective with the optimization of quantile perturbed losses. We leave this for future work.
>
> [Li et al. 2023] On Tilted Losses in Machine Learning: Theory and Applications, JMLR 2023.

---

> ### Comment · Reviewer_aGF9 · 2024-11-25
>
> Thanks for the detailed reply. I increase my score to 5.
>
> Please ensure that experiments involving training on ImageNet-1k are included in your next version. Thanks!

---

> > ### Author Response · Authors · 2024-11-28
> >
> > Thank you for considering our responses and increasing the score. We will include ImageNet-1k results in the next version.

---

### Official Review · Reviewer_uH13 · 2024-11-03

**Soundness:** 3
**Presentation:** 3
**Contribution:** 3
**Rating:** 8
**Confidence:** 4

**Summary:**

This paper presents an approach named Tilted Sharpness-Aware Minimization (TSAM), which extends the Sharpness-Aware Minimization (SAM) method to improve generalization in overparameterized machine learning models. SAM optimizes model parameters by focusing on minimizing the maximum loss in a neighborhood, aiming to find flat minima that enhance generalization. However, SAM's approach can be computationally challenging and overly focused on the worst-case local solutions.

TSAM addresses these issues by introducing a "tilt" parameter formulation $t$, which allows for reweighting local minima, focusing on flatter solutions that could generalize better. The authors demonstrate that as the tilt parameter increases, TSAM smooths the optimization landscape, favoring flatter minima. To effectively solve the TSAM objective, the authors propose a novel Hamiltonian Monte Carlo-inspired algorithm to sample and estimate gradients efficiently. Empirical evaluations across image and text tasks show TSAM’s superiority in test performance over SAM and ERM, including fine-tuning ResNet18 and Vision Transformers on CIFAR-100, ImageNet, and DTD datasets, and fine-tuning DistilBert model on the GLUE benchmark

**Strengths:**

- TSAM’s approach to generalization via tilting provides a novel and flexible framework, allowing it to optimize for both worst-case and average-case scenarios, unlike SAM’s rigid min-max formulation.Theoretical proofs and empirical results support that TSAM achieves flatter minima than SAM, correlating with improved generalization on test data.

- TSAM is tested on a diverse range of tasks and models (e.g., CNNs, transformers, and GLUE benchmark tasks), demonstrating its robustness and adaptability to different domains.

- By using a Hamiltonian Monte Carlo method to sample neighborhood perturbations, TSAM optimizes for flatter minima more efficiently than naive sampling would allow, enabling practical deployment in high-dimensional optimization tasks.

**Weaknesses:**

- The algorithm introduces additional parameters (e.g., tilt parameter $t$ and sampling frequency), requiring careful tuning, which could be challenging in resource-constrained environments.

- Although TSAM is more efficient than a straightforward min-max approach, it still requires multiple gradient evaluations per iteration. Can the authors analyze the runtime complexity of the optimization algorithm? Moreover, it would be better to provide a comparison of running time between the baselines in the paper.

**Questions:**

How can one tune the hyper-parameter $t$? It would be better to provide a clear method/intuition for selecting an optimal $t$.

---

> ### Author Response · Authors · 2024-11-20
> **response**
>
> We are grateful for the insightful reviews, and the acknowledgement of the theoretical and empirical contributions of our work.
>
> **[Runtime comparisons]** We note that on the image datasets of Table 1, we compare the performance of various algorithms under the same computational budget. Though TSAM requires more gradient computation per iteration, it still outperforms baselines with fewer total iterations. We report the exact runtime numbers for TSAM and baselines in Table 6 of the revision. In terms of optimization complexity, if we assume we can get an unbiased estimator of the tilted gradients at each iteration, the convergence speed of the optimization algorithm is $O(1/T)$ for strongly convex and smooth functions, following standard arguments in optimization. TSAM is also easily parallelizable by sampling perturbations at the same time; we will make such implementation optimization in the next version.
>
> **[Tuning t]** While we usually need to tune $t$ manually, one goal of this work is to provide some high-level intuitions around effects of $t$. For instance, we discuss smoothness, convexity, flatness, and generalization as functions of t in Section 3. It is an interesting future direction to explore how to automatically optimize $t$ in an end-to-end way. One high-level idea is to take inspirations from previous works (e.g., Eq. (65) in Li et al. (2023)) and connect the TSAM objective with the optimization of quantile perturbed losses. It is thus possible to reformulate the problem, taking $t$ as an optimization variable as opposed to a hyperparameter. We leave this for future work. Thanks for bringing this up!
>
> [Li et al., 2023] On Tilted Losses in Machine Learning: Theory and Applications, JMLR 2023.

---

> > ### Comment · Reviewer_uH13 · 2024-11-24
> >
> > Thanks so much for the authors' responses! I would like to maintain my acceptance to the paper.
> >
> > The discussion about the runtime and the tuning of $t$ is very interesting. It would be better to make them explicit in the paper, especially the fact that "Though TSAM requires more gradient computation per iteration, it still outperforms baselines with fewer total iterations".
> >
> > Moreover, there are recent developments in optimizing a noise-injected loss, which regularizes the trace of Hessian of the network weights. It would be interesting to discuss the connection between this paper to these related methods, such as:
> > - Noise Stability Optimization for Finding Flat Minima: A Hessian-based Regularization Approach. TMLR 2024

---

> > > ### Author Response · Authors · 2024-11-25
> > > **response**
> > >
> > > Thanks for the continued positive assessment of our work! We have made the runtime arguments more explicit throughout the paper by pointing out that we compare with the baselines under the same computational budget.
> > >
> > > Thanks for sharing this interesting recent work. In particular, our TSAM objective with $t$=0 is very similar to the proposed noise-injection objective (Eq. (1) in the paper). The proposed algorithm therein can cancel out the first-order gradient terms and explicitly turn the objective into minimizing mean plus trace of Hessian, which aligns with the argument that the TSAM objective can lead to flatter solutions. We have added the discussion to the revision (introduction and Section 3.2).

---

### Official Review · Reviewer_wkWd · 2024-11-04

**Soundness:** 3
**Presentation:** 3
**Contribution:** 3
**Rating:** 6
**Confidence:** 4

**Summary:**

The paper introduces a new version of the sharpness-aware minimization algorithm (SAM) called tilted SAM (TSAM).
TSAM uses exponential tilting to smooth the objective. TSAM comes with a temperature parameter t.
 As t approaches infinity TSAM converges to classic SAM.
The paper shows exponential tilting leads to an easier optimization problem while still favoring flat solutions.
TSAM's theoretical analysis shows that it enjoys good generalization properties.
Moreover, the numerical experiments show TSAM performs superior to other SAM-style algorithms on several benchmark tasks.

**Strengths:**

**Clarity**

The paper is well-presented and easy to read.

**Theory**

The paper provides good theoretical guarantees for the algorithm, establishing that the solution found by the algorithm generalizes well under standard hypotheses.

**Experiments**

TSAM performs better than existing SAM-like approaches on standard ML tasks.
Moreover, the competitors are tuned, which is good.

**Overall**

This is a solid paper and could be of interest to certain subcommunities working on deep learning theory and optimization.

**Weaknesses:**

**Technical Contribution**

The argument for establishing the main result, Theorem 2, seems standard. It consists of a standard application of Hoeffding's inequality combined with a technical lemma proved by Aminian et al. (2024).

**Numerical Experiments**

The paper only reports performance over epochs and not wall clock time. It should report both; if the proposed algorithm runs significantly slower than existing methods, that's problematic.

For the HMC portion of the algorithm, to generate $\epsilon$, the authors mention they set $N=1$ for computational efficiency.
Given the experimental results, this approximation seems to work.
Nevertheless, it would have been nice to have some ablation in the appendix with larger values of $N$ to show the impact increasing this parameter has on performance.
This would help validate the setting of $N=1$ and show if there is any tradeoff between speed and performance in setting $N$.

**Questions:**

1. Could you add wall clock times in the revision?
2. Could you add an ablation to the appendix for the $N$ parameter.

---

> ### Author Response · Authors · 2024-11-20
> **response**
>
> We thank the reviewer for the positive review of our work.
>
> **[Wall clock times]** Though we did not report the exact runtime of the experiments, we compare different methods under the same computation time in Table 1 (our main result), including expensive versions of SAM  (ESAM1 and ESAM2) and letting other sharpness-aware variants (PGN and RSAM) run longer than TSAM. We show that under the same runtime, TSAM outperforms other algorithms (expensive SAM and other SAM variants). We have added the exact wall clock time in the revision (Table 6, Appendix 4.2) and highlighted them in blue.
>
> **[Effects of N]** Our empirical results in the original submission are based on running $N$=1 steps in HMC (Algorithm 1 and Algorithm 2). Per the reviewer’s suggestion, we still report the performance with $N$>1 in Table 4, Appendix C.2 of the revision. We assume we accept all the generated epsilons for simplicity and easier comparisons. For different $N$’s, we sample the same number of perturbations. The result is also copied in the table below. We observe diminishing returns when increasing $N$.
>
> |      | CIFAR100 | DTD |
> |-----------|--------------------------|----------|
> | N=1 | 0.7740 | 0.6882 |
> |N=2 | 0.7745 | 0.6883 |
> |N=3| 0.7752 | 0.6880 |
>
>
> **[Others]** Thanks for the reviewer’s comments on “...this paper provides good theoretical guarantees…”. Regarding the proof of Theorem 2, we agree that the proof is not technically complicated, which we view as a strength, since we arrive at meaningful results with clean proofs built upon previous works. Theorem 2 implies that there exists an optimal $t$ that minimizes the upper bound of the generalization error defined as linear population risk. Additionally, we provide various major theoretical arguments in Section 3  (e.g., smoothness, convexity, and flatness of solutions) in addition to the theorem discussed here to help understand the properties of TSAM.
>
>
> ​​

---

> > ### Comment · Reviewer_wkWd · 2024-11-28
> > **Thank you for your reply.**
> >
> > Hi, I want to thank the authors for responding to my comments/concerns. I appreciate the ablation of the $N$ parameter and the addition of wall-clock times to the supplement.
> >
> > At this time, I will maintain my score and recommend weak acceptance.

---

### Official Review · Reviewer_1jA6 · 2024-11-04

**Soundness:** 2
**Presentation:** 3
**Contribution:** 3
**Rating:** 6
**Confidence:** 4

**Summary:**

Inspired by the idea of exponential tilting in probability and statistics, this paper introduces a generalized and smoothed variant of SAM, termed TSAM. The authors highlight two key attributes of TSAM: smoothness and preference for flatter minima. They further develop an algorithm to practically solve TSAM. Finally, they verify the generalization improvement of TSAM over SAM and ERM via extensive experiments.

**Strengths:**

1. The idea of reweighting multiple points in the neighborhoods of the local minima instead of focusing on the single worst solution is intuitive.

2. The paper is well-written and supports its claims through theory and comprehensive experiments.

**Weaknesses:**

1. There seem to be conflicting explanations for why TSAM outperforms SAM regarding generalization. On one side, the authors suggest that TSAM improves generalization by considering a broader range of local solutions rather than concentrating solely on the worst-case scenarios, indicating that TSAM's objective function is inherently superior for identifying solutions with better generalization. Conversely, TSAM is said to favor flatter minima as $t$ increases. As $t$ approaches infinity, TSAM converges to SAM, suggesting that SAM might ideally be the best at finding flatter minima, if not for practical optimization challenges. Thus, TSAM's enhanced performance in practice may stem from its balanced trade-off between optimizing for flatter minima and maintaining optimization efficiency.

2. The generalization bounds presented in Section 3.3 are confusing regarding the message they aim to convey. The authors seek to demonstrate that TSAM has smaller upper bounds on generalization error compared to SAM and ERM. However, in their theoretical framework, the bound for SAM is worse than that for ERM, which contradicts the superior generalization performance of SAM in practice. This contradiction undermines the validity of the comparative advantage attributed to TSAM over both SAM and ERM.

3. The improvement of TSAM over SAM and ERM is marginal in the transformer experiments (the second table in Table 1). Moreover, TSAM requires more computation than SAM and ERM, thus raising the question: if variants such as ESAM1 or ESAM2 were to be tested, would the marginal improvements provided by TSAM diminish further?

**Questions:**

See weaknesses

---

> ### Author Response · Authors · 2024-11-20
> **response**
>
> **[Flatness and generalization of TSAM solutions]** We agree with the reviewer that TSAM outperforms SAM due to the combined effects of encouraging flat minima and the smoothness of the objective. The relations between flatness and generalization remains an interesting open problem (e.g., Wen et al., 2023). In Theorem 1, we comment on the flatness property of TSAM for generalization linear models, which by itself does not directly prove that TSAM generalizes better than SAM. Regarding the generalization bound in Theorem 2, the upper bound of $E_Z[l(\theta, Z)]$ (the linear population risk that we are concerned about) converges to the classic generalization result of ERM as $t$ converges to 0. Hence, our result is relatively tight for a bounded $t$, implying the superiority of TSAM over ERM. We agree that the bound in its current form does not explain the superiority of SAM over ERM, as the optimal $t$ that minimizes the upper bound is not $\infty$. We have revised the corresponding paragraph in the revision to avoid confusion.
>
> [Wen et al., 2023] Sharpness Minimization Algorithms Do Not Only Minimize Sharpness To Achieve Better Generalization, NeurIPS 2023.
>
> **[Empirical improvements]** Thanks for the suggestion of evaluating other baselines on the language tasks. We note that compared with numbers reported in previous works (e.g., Bahri et al., Table 2), our improvements are on a similar scale (if we convert the results in our paper to the form of percentage). We observe that on bert experiments, finetuning can be done typically within a small number of epochs (e.g., 5 to 10 depending on the dataset in the Glue benchmark). If we continue to finetune using SAM (ESAM1), the performance will not improve over SAM. For the ESAM1 and ESAM2 baselines, the results on the Glue benchmark are as follows. TSAM still demonstrates higher scores. We have also added these results in the second table of Table 1.
>
> |     | **CoLA** | **WNLI** | **SST-2** | **MNLI** | **QNLI** | **RTE** | **MRPC** | **QQP** | **STSB** | **Average** |
> |-----------|--------------------------|----------|-----------|----------|----------|----------|----------|----------|----------------------------|------------|
> | SAM       | 0.52 /0.8048     | 0.5634      | 0.9174     | 0.811    | 0.8642     | 0.5884     | 0.8529     | 0.8771 | 0.870/0.865           | 0.7756 |
> | ESAM1       | 0.52/0.8044       | 0.5634   | 0.9163  | 0.811 | 0.8618     | 0.5902 | 0.8531    | 0.8769 | 0.871/0.867                       | 0.7759       |
> | ESAM2    | 0.52 /0.8053     | 0.5634      | 0.9163     | 0.812     | 0.8628     | 0.5925     | 0.8580     | 0.8747 | 0.868/0.865             | 0.7763       |
> | TSAM     | 0.5/0.8081     | 0.5634      | 0.9186     | 0.811    | 0.8781     | 0.6065     | 0.8505     | 0.8877 | 0.871/0.866                       | 0.7801|
>
>
> [Bahri et al., 2022] Sharpness-Aware Minimization Improves Language Model Generalization, ACL 2022.

---

> ### Comment · Reviewer_1jA6 · 2024-11-30
> **Thank you for your response and I will keep my positive score.**
>
> Thank you for clarifying the advantages of TSAM over SAM and for doing the additional experiments on ESAM. These new experiments addressed my concerns regarding the effectiveness of TSAM. However, as noted in Weakness 2 and by other reviewers, TSAM incurs more computational overhead compared to SAM. Given this limitation, I have decided not to raise my score, but I still lean towards acceptance.

---

### Meta-Review · Area_Chair_8WV6 · 2024-12-19

**Metareview:**

The paper introduces Tilted Sharpness-Aware Minimization, a generalization of SAM that smooths the objective using exponential tilting. While the reviewers acknowledged the theoretical exploration and the intuitive motivation behind TSAM, they raised significant concerns that remain unresolved.

Key concerns include:

1) Marginal Empirical Improvement: Although TSAM shows improvements over SAM and ESAM on standard benchmarks, the gains (1-2%) are considered minor, particularly in light of the substantial computational overhead.
2) Computational Cost: TSAM requires 3-5 times the computational time and memory of SAM, making it impractical for large-scale datasets.
3) Theoretical Contributions: While the analysis supports TSAM’s preference for flatter minima, reviewers noted that the bounds rely on standard results and do not provide new insights into the relationship between flatness and generalization.
4) Hyperparameter Sensitivity: The tilt parameter requires careful tuning, further increasing computational costs.

The authors provided clarifications during the rebuttal, but the reviewers felt the issues regarding practical utility, computational efficiency, and theoretical significance were not fully resolved. Consequently, I recommend rejection.

**Additional Comments On Reviewer Discussion:**

During the discussion with the AC, reviewers raised concerns about the computational overhead of Tilted SAM, the marginal empirical improvements compared to SAM and other variants, and the sensitivity of the tilt hyperparameter. While the authors provided clarifications on experimental settings and performance gains during the rebuttal, some reviewers were not fully convinced that the improvements justified the additional computational cost. The theoretical contributions, though sound, were also deemed incremental. Following a thorough discussion between the AC and the reviewers, I propose to reject the paper based on these concerns.

---

### Decision · Program_Chairs · 2025-01-22

Reject